# Boosting a practical Li-CO$_2$ battery through dimerization reaction based on solid redox mediator

Wei Li[1], Menghang Zhang[1], Xinyi Sun[1], Chuanchao Sheng[1], Xiaowei Mu[1], Lei Wang[1], Ping He ☉[1] ✉ & Haoshen Zhou ☉[1] ✉

Li-CO$_2$ batteries offer a promising avenue for converting greenhouse gases into electricity. However, the inherent challenge of direct electrocatalytic reduction of inert CO$_2$ often results in the formation of Li$_2$CO$_3$, causing a dip in output voltage and energy efficiency. Our innovative approach involves solid redox mediators, affixed to the cathode via a Cu(II) coordination compound of benzene-1,3,5-tricarboxylic acid. This technique effectively circumvents the shuttle effect and sluggish kinetics associated with soluble redox mediators. Results show that the electrochemically reduced Cu(I) solid redox mediator efficiently captures CO$_2$, facilitating Li$_2$C$_2$O$_4$ formation through a dimerization reaction involving a dimeric oxalate intermediate. The Li-CO$_2$ battery employing the Cu(II) solid redox mediator boasts a higher discharge voltage of 2.8 V, a lower charge potential of 3.7 V, and superior cycling performance over 400 cycles. Simultaneously, the successful development of a Li-CO$_2$ pouch battery propels metal-CO$_2$ batteries closer to practical application.

Recently, the reversible Li-CO$_2$ battery has gained significant attention as a promising and innovative solution for next-generation energy storage systems[1–5], boasting a high theoretical energy density of approximately 1876 Wh kg$^{-1}$. Through the typical reversible reaction (as described in Eq. 1), this battery has the potential to mitigate the impact of global warming by reducing the use of fossil resources[6,7] and increasing the capture of CO$_2$. In addition, the Li-CO$_2$ battery has promising applications in the space industry and specifically in the exploration of Mars, where the atmospheric CO$_2$ content is as high as 96 percent[8,9].

$$4Li^+ + 3CO_2 + 4e^- \leftrightarrow 2Li_2CO_3 + C \qquad (1)$$

However, in reality, there are still many technical challenges to be overcome for its widespread implementation. Firstly, the direct electroreduction of CO$_2$ is challenging due to its inherent electrochemical inertness, which results in a low discharge voltage (<2.0 V), and even lower than 1.5 V, leading to a significant drop in the actual specific

energy of the battery[10–12]. Secondly, Li$_2$CO$_3$, the main discharge product of the reaction, is a wide-bandgap insulator, which leads to slow kinetics and high voltage platforms (>4.3 V) during the charging process[13–15]. This large overpotential significantly decreases the round-trip efficiency of the battery. Additionally, the irreversible decomposition of Li$_2$CO$_3$ can cause "sudden death" of the Li-CO$_2$ battery (Eq. 2)[16,17].

$$2Li_2CO_3 \rightarrow 2CO_2 + O_2^-(^1O_2) + 4Li^+ + 3e^- \qquad (2)$$

Recent research has found that Li$_2$C$_2$O$_4$ as the final discharge product can reduce the charge potential to 3.8 V using Mo$_2$C as cathode catalyst in Li-CO$_2$ battery[18,19]. However, these catalysts are not effective in stabilizing the Li$_2$C$_2$O$_4$ product, leading to limited cycle life of only up to 40 cycles, which is far from meeting the requirements for long-term operation. To address the challenges of low discharge potential and large charge polarization, some soluble redox mediators (soluble RMs) have been introduced in Li-CO$_2$ batteries, which has

[1]Center of Energy Storage Materials & Technology, College of Engineering and Applied Sciences, Jiangsu Key Laboratory of Artificial Functional Materials, National Laboratory of Solid-State Microstructures and Collaborative Innovation Center of Advanced Microstructures, Nanjing University, Nanjing 210093, PR China. ✉e-mail: pinghe@nju.edu.cn; hszhou@nju.edu.cn

previously been widely used in Li-O$_2$ batteries. Although certain some soluble RMs enhance electrochemical properties such as round-trip efficiency, discharge capacity, and so on, Li-CO$_2$ batteries based on these RMs always exhibit unsatisfactory reversibility (often fewer than 70 cycles)[20–22]. However, in addition to poor solubility, there still are some critical issues, namely the shuttle effect and sluggish kinetics, which severely limit the practical application of soluble RM-based rechargeable batteries[23]. The shuttle effect, that is, the undesirable diffusion of species between electrodes, which can lead to adverse side reactions, such as soluble RM decomposition and Li-metal deterioration. To this end, Li-deficient Li$_x$FePO$_4$ ($x = 0.9$) as the anode has been employed in previously reported researches[23,24]. However, this is not a true Li metal-CO$_2$ battery. Meanwhile, due to the existence of concentration polarization of soluble RM, it is difficult to tackle the problem of sluggish kinetics at large current. This makes soluble RM unable to diffuse to the cathode surface in time at large current, thus affecting the reduction reaction of CO$_2$ and the reversibility of battery, leading to the application of soluble RM in Li metal-CO$_2$ batteries is not practical[25,26].

In contrast to conventional soluble RMs, we put forward a strategy of using solid redox mediators (solid RMs) fixed and anchored on the cathode of Li-CO$_2$ batteries. This approach effectively circumvents the shuttle effect and kinetics issues of soluble RMs in the electrolytes. It is worth mentioning that solid RMs are also different from the previously reported solid catalysts. Solid RMs can replace CO$_2$ molecules to undergo electrochemical reactions on the electrode surface. The discharge reaction of the Li-CO$_2$ batteries has changed from the electrochemical reduction of CO$_2$ to the electrochemical reduction of solid RMs. To improve discharge voltage and reduce charge polarization, an effective solid RMs should possess unique chemical binding ability with CO$_2$, along with the following characteristics: reversible redox couples, good electronic conductivity, high specific surface area, ample active sites, and low preparation cost.

In this work, we present a solid redox mediator Cu(II) metal-organic framework (MOF) of benzene-1,3,5-tricarboxylic acid (denoted as solid RM(II)-BTC, II represents the valence state of copper ion in RM(II)-BTC). Solid RM(II)-BTC is a microporous MOF composed of metal ions (Cu$^{2+}$) and organic ligands (benzene-1,3,5-tricarboxylic acid, BTC) with 9 Å pores that can absorb large amounts of CO$_2$[27]. So, solid RM(II)-BTC may be versatile in the adsorption and catalytic reduction of CO$_2$ in organic electrolyte. This can effectively mitigate the shuttle effect and sluggish kinetics associated with conventional soluble RMs (Fig. 1a), while also reducing the charge/discharge potential gap via the Li$_2$C$_2$O$_4$ pathway. Specifically, during discharge, solid RM(II)-BTC

anchored on the surface of CNTs is first reduced to RM(I)-BTC. Then, RM(I)-BTC reacts with adsorbed CO$_2$ to produce C$_2$O$_4^{2-}$ and RM(II)-BTC, along with the final product of Li$_2$C$_2$O$_4$, through the catalyzation of reversible Cu(II)/Cu(I) redox coupling (Fig. 1b). This study demonstrates the superior electrochemical performance of Li-CO$_2$ batteries loaded with the solid RM(II)-BTC. We analyzed and confirmed the electrocatalytic reduction process of CO$_2$ by solid RM(II)-BTC during the discharge process, as well as the corresponding charge reaction pathway of the battery. This solid RM(II)-BTC exhibits the similar efficacy as traditional soluble RMs, but its solid phase prevents the shuttle effect and consumption on the negative electrode. Compared with traditional soluble RMs, this solid RM(II)-BTC can support the lithium-metal negative electrode, making it more practical. Meanwhile, concentration polarization of soluble RMs is avoided, and the poor dynamics issue under high current is also considerably addressed. The proposal of the solid redox mediator lays the foundation for the practical application of Li-CO$_2$ batteries.

## Results and discussion
### Synthesis and characterization of solid RM(II)-BTC
The hierarchical porous solid RM(II)-BTC is synthesized using a solid precursor-assisted confinement conversion method. The synthesizing process for the solid RM(II)-BTC composite material is shown in the Supplementary Fig. 1 and described in detail in the Experimental Section[28,29]. Single-walled carbon nanotubes (CNTs) with small diameters and negatively charged surfaces easily assemble with highly positively charged copper hydroxide nanostrands through simple mixing. The mixed solution is then filtered onto a porous substrate to obtain the CNTs/CHNs composite material[30,31], which is then immersed in a pre-prepared organic ligand solution (BTC) for 1 h at room temperature to produce the solid RM(II)-BTC/CNTs composite material. As shown in Fig. 2a−c, the material's large specific surface area (1650 m$^2$ g$^{-1}$) and appropriate pore size (9 Å), combined with durable metal sites, facilitate strong CO$_2$ adsorption (9.02 mmol g$^{-1}$)[27]. It is worth noting that "RM(II)-BTC" is merely a representation used in this paper, which represents a structural unit containing two Cu atoms in the MOF crystal, as shown in Figs. 1b and 2c. To confirm the accuracy of the composite structure, a series of characterization techniques are employed. X-ray diffraction (XRD) is used to determine the crystalline phase composition of the solid RM(II)-BTC material, as shown in Fig. 2d. The characteristic peaks at 6.5°, 9.3°, 11.5°, 13.4°, 14.6°, 16.3°, 17.4°, 18.9° and 20.1° correspond to the diffractions of the (220), (220), (222), (400), (331), (333), (511), (600) and (620) lattice planes of the solid RM(II)-BTC structure, respectively[32]. FT-IR spectroscopy (Fig. 2e)

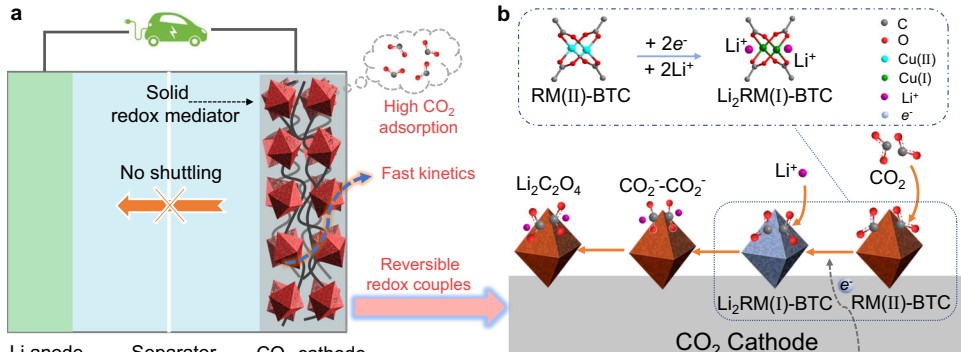

**Fig. 1 | Schematic and mechanism diagram of Li-CO$_2$ battery with RM(II)-BTC cathode. a** Schematic of Li-CO$_2$ battery with solid RM(II)-BTC cathode. RM(II)-BTC can inhibit the shuttle effect since it is fixed at the cathode as a solid state and exhibit high CO$_2$ adsorption, fast kinetics and reversible redox couples. **b** Solid RM(II)-BTC-mediated reaction mechanism towards Li$_2$C$_2$O$_4$ route. RM(II)-BTC is electrochemically reduced to Li$_2$RM(I)-BTC, which immediately reacts with CO$_2$ to form Li$_2$RM(II)-BTC-2CO$_2^-$. Finally, discharge product Li$_2$C$_2$O$_4$ and origin RM(II)-BTC are obtained. The frame with dash-dotted line reflect that after receiving electrons and incorporating lithium ions, RM(II)-BTC can transfer to a compound of lithium and RM(I)-BTC. The gray, red, cyan, green, purple and blue balls represent C, O, Cu(II), Cu(I), Li$^+$ and e$^-$, respectively.

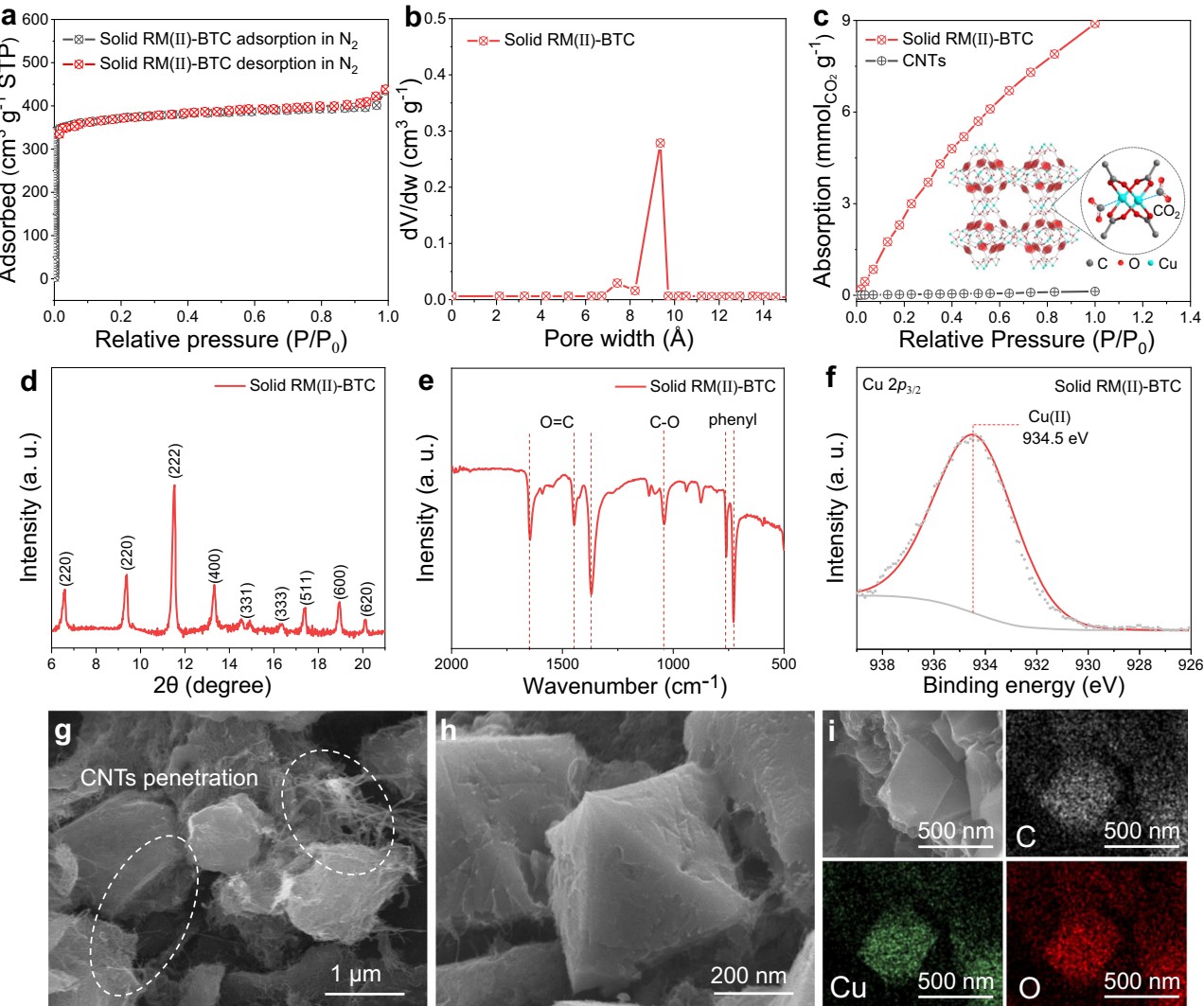

**Fig. 2 | Schematics illustrating the properties and characterization methods of solid RM(II)-BTC cathode. a** $N_2$ adsorption-desorption isotherms of solid RM(II)-BTC cathode at 77 K. **b** Pore size distributions of solid RM(II)-BTC cathode. **c** $CO_2$ adsorption isotherms of solid RM(II)-BTC and CNTs at 298 K. Insets: the unit cell structure diagram of solid RM(II)-BTC that is a structural unit containing two Cu atoms. The gray, red, and cyan balls represent C, O, and Cu(II), respectively. **d** XRD pattern, (**e**) FT-IR spectroscopy, (**f**) XPS survey, (**g, h**) SEM images and (**i**) SEM image and EDS spectroscopy of solid RM(II)-BTC cathode.

of the solid RM(II)-BTC shows multiple characteristic peaks around 1684, 1445, 1370, 1041, 760, and 730 $cm^{-1}$, which are attributed to the stretching vibrations of C = O, C-O, and phenyl.

In addition, the presence of Cu(II) can be identified by the peak at 936.5 eV in the $Cu2p_{3/2}$ spectrum of solid RM(II)-BTC through X-ray photoelectron spectroscopy (XPS), as shown in Fig. 2f[33,34]. The prepared composite material has a three-dimension network structure with evenly dispersed solid RM(II)-BTC nanoparticles (0.5–2 μm) within the CNTs framework, as seen in the scanning electron microscopy (SEM) images (Fig. 2g, h). The permeable three-dimensional conductive network of CNTs significantly contributes to electronic conduction both between and within the MOF (3 S $cm^{-1}$). The element mapping images from energy-dispersive X-ray spectroscopy (EDS) also show that the elements Cu, C, and O are evenly distributed on the surface of the solid RM(II)-BTC particles (Fig. 2i). Thus, all characterization techniques demonstrate that we have successfully synthesized the solid RM(II)-BTC. Additionally, the mass content of solid RM(II)-BTC and CNTs in the RM(II)-BTC/CNTs composite material is determined through thermogravimetric analysis (TGA) and is found to be 28.7% and 71.3%, respectively (Supplementary Fig. 2).

## Electrochemical performance of Li-CO₂ battery catalyzed by solid RM(II)-BTC

The Li-CO₂ batteries are assembled and investigated to thoroughly evaluate the catalytic activity of solid RM(II)-BTC cathode toward the $CO_2$ reduction reaction (CO₂RR) and $CO_2$ evolution reaction (CO₂ER). The cyclic voltammetry tests are performed with a limited voltage range of 2.0–4.5 V at a scanning rate of 0.1 mV $s^{-1}$. Under Ar atmosphere, as shown in Fig. 3a, solid RM(II)-BTC cathode exhibits distinct redox peaks at 2.81 V ($E_{c1}$) and 3.34 V($E_{a1}$), respectively, indicating that the reversible reduction and oxidation of Cu(II)/Cu(I) may occur in solid RM(II)-BTC[23,35]. The individual redox potentials of active metal centers in complex compounds are influenced by the properties and structure of ligands[36–38]. Similarly, in the CO₂ atmosphere, the presence of a reduction peak (2.81 V, $E_{c2}$) at the same position of CV curve indicates that the same reduction process occurs in the solid RM(II)-BTC. In addition, compared with the oxidation peak at 3.34 V ($E_{a1}$) in Ar, two new oxidation peaks appear at around 3.40 V ($E_{a2}$) and 3.70 V ($E_{a2}'$) in CO₂. The peak at 3.40 V ($E_{a2}$) corresponds to the oxidation of trace Cu(I) ($E_{c2}$), while the peak at 3.70 V ($E_{a2}'$) is more likely attributed to the decomposition of the discharge product. However, the oxidation peaks in Ar ($E_{a1}$) ad CO₂ ($E_{a2}$) gas are different and shows a shift in CO₂

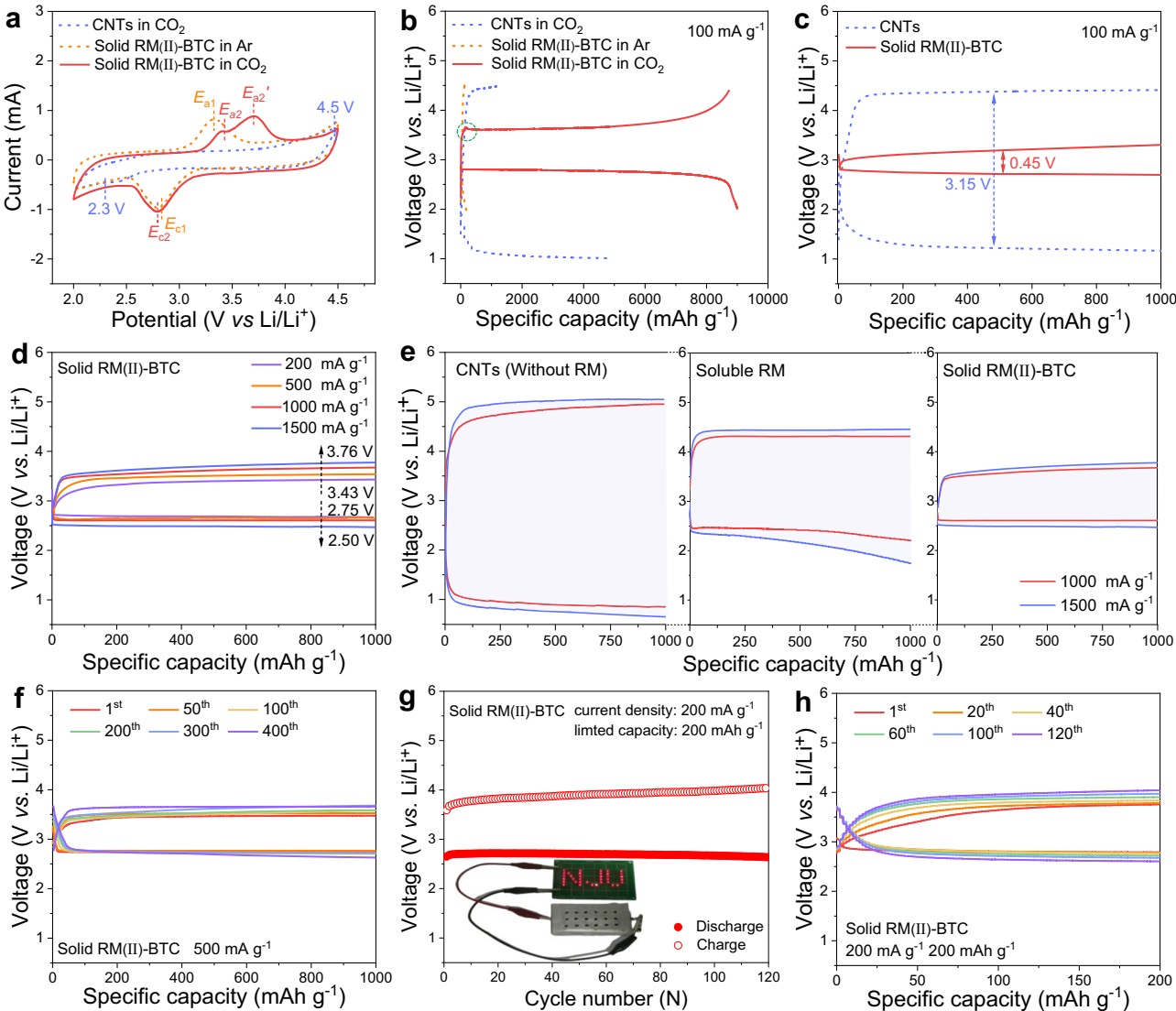

**Fig. 3 | Electrochemical performance of Li-CO₂ batteries with CNTs and solid RM(II)-BTC cathode. a** The CV plots of CNTs and solid RM(II)-BTC cathode in Ar and CO₂ atmosphere. **b** Full deep discharge/charge profiles of CNTs and solid RM(II)-BTC cathode under Ar and CO₂ at a current density of 100 mA g⁻¹. **c** Comparison of the galvanostatic discharge/charge profiles of CNTs and solid RM(II)-BTC cathode with a fixed capacity of 1000 mAh g⁻¹ at 100 mA g⁻¹. **d** Comparison of the galvanostatic discharge/charge profiles of solid RM(II)-BTC at various current densities. **e** Comparison of the galvanostatic discharge/charge profiles of CNTs (without RM), soluble RM and solid RM(II)-BTC at high current densities (1000 and 1500 mA g⁻¹). **f** Cycling behavior of solid RM(II)-BTC cathode for the selected cycles under CO₂ at a current density of 500 mA g⁻¹ and curtailing capacity of 1000 mAh g⁻¹. **g** Cycling performance for pouch Li-CO₂ batteries with solid RM(II)-BTC cathode at 200 mA g⁻¹ within 200 mAh g⁻¹, along with red LED lights powered by this device in CO₂ glove box as insets. **h** Cycling behavior of the fabricated pouch battery of solid RM(II)-BTC cathode for the selected cycles under CO₂ at a current of 200 mA g⁻¹ and curtailing capacity of 200 mAh g⁻¹.

with respect to Ar. This is owing to the presence of carbon dioxide discharge products on the electrode surface with low electrical conductivity (Li₂C₂O₄), resulting in enhanced electrode polarization. In contrast, a pure CNTs cathode displays a typical response for carbon-based cathodes under CO₂, with a couple of cathodic/anodic peaks at 2.3 V and 4.5 V, respectively[39,40]. By comparing these redox peaks, it appears that the reaction path of Li-CO₂ battery is altered in the presence of solid RM(II)-BTC.

To investigate the catalytic performance of the solid RM(II)-BTC cathode, full discharge-charge tests were performed on Li-CO₂ batteries with a limited voltage range of 2.0–4.5 V at a current density of 100 mA g⁻¹. As shown in Fig. 3b, the solid RM(II)-BTC cathode can provide a larger initial discharge specific capacity of up to 9040 mAh g⁻¹ and a reversible capacity as high as 8747 mAh g⁻¹. It is worth emphasizing that the discharge product can be decomposed below 3.7 V with a higher discharge plateau of 2.8 V. The control

group of CNTs cathode only delivers a discharge capacity of 4680 mAh g⁻¹ and a recharge capacity of 1150 mAh g⁻¹ under CO₂, with a lower discharge potential of 1.1 V and a higher charge potential over 4.45 V.

These results suggest that the solid RM(II)-BTC cathode has a significant influence on increasing discharge capacity and decreasing charge overpotential. An inconspicuous slope appears at the start of the charging curve in the battery containing solid RM(II)-BTC, possibly due to the oxidation of Cu(I) to Cu(II) after deep discharge[24]. As shown in Supplementary Fig. 3, the deconvoluted Cu 2p spectrum of cathode after deep discharge shows peaks assigned to Cu(I) and Cu(II). Conversely, in an Ar environment, the discharge/charge capacity drops significantly even when solid RM(II)-BTC is present. The capacity arising from the redox of the Cu(II)/Cu(I) couple in solid RM(II)-BTC in Ar is about 95 mAh g⁻¹, which is close to its theoretical specific capacity of 121 mAh g⁻¹ based on the reversible reaction (Eq. 3) (Supplementary

Fig. 4)[41]. Although the discharge capacity of solid RM(II)-BTC itself in an Ar environment is completely negligible, when it interacts with $CO_2$, its capacity increases dramatically. This suggests that the capacity contribution comes from the combined action of solid RM(II)-BTC and $CO_2$, rather than the solid RM(II)-BTC material alone. Given this, the structural stability of the materials is crucial for the solid RM(II)-BTC to play a long-term role in the system. Subsequently, in-situ XRD results also directly confirmed that the structure of solid RM(II)-BTC remains stable during the charging and discharging operations in an Ar atmosphere. At the same time, the non-attenuation of the capacity also indicates that it can continue to act as an solid RM in the electrochemical reduction of $CO_2$ through the Cu(II)/Cu(I) reversible redox coupling (Supplementary Fig. 5a, b).

$$2Li^+ + 2e^- + RM(II) - BTC \leftrightarrow Li_2RM(I) - BTC \quad (3)$$

To better illustrate the significant decrease in overpotential, we compare the galvanostatic discharge/charge experiment with a limited capacity of 1000 mAh g$^{-1}$ between the CNTs and solid RM(II)-BTC cathodes, as shown in Fig. 3c. It is evident that the pure CNTs cathode exhibits a much larger charge/discharge potential gap of 3.15 V, which is nearly seven times larger than that of solid RM(II)-BTC cathode ($\approx 0.45$ V).

Furthermore, rate capability is an important indicator for reaction kinetics. As shown in Fig. 3d, with the increase in current density from 200 to 500, 1000 and 1500 mA g$^{-1}$ on solid RM(II)-BTC cathode, the discharge voltage shows a slight decrease from 2.75 to 2.50 V, and the charge termination voltage rising from 3.43 to 3.76 V. In comparison, pure CNTs cathode (without RM) suffers from much severe voltage polarization at every applied current density (Supplementary Fig. 6). Importantly, even compared with the soluble RM just reported by our group, solid RM(II)-BTC also shows great advantages in rate capability, especially at high current densities (1000 and 1500 mA g$^{-1}$) (Fig. 3e). Electrochemical impedance spectroscopy (EIS) technique can be used to further analyze the kinetics of discharge processes at high discharge current densities with a fixed capacity (1000 mAh g$^{-1}$). The electrochemical impedances during discharging process for various catalysts, including CNTs (without RM), soluble RM and solid RM(II)-BTC, are compared in Supplementary Fig. 7, and an equivalent circuit is used to fit the EIS data (Supplementary Table 1)[42,43]. The circuit elements are defined as follows, $R_s$ reflects the ohmic resistance that comprises contributions from the electrolyte, electrodes, current leads, and so on. $R_{sei}$ denotes the resistance of solid electrolyte interfacial layers on the air electrode, whereas $C_{sei}$ represents its capacitance. Similarly, $R_{ct}$ and $C_{dl}$ are attributable to charge-transfer resistance and double-layer capacitance, respectively. $W$ represents the Warburg impedance, that arises from a diffusion process. The total impedance ($R_t$) in the entire process, namely, the asymptotic limit of the real section of the impedance spectroscopy, grows with increasing current density (1000 to 1500 mA g$^{-1}$) for CNTs (670 to 965 Ω) and soluble RM (260 to 420 Ω). However, the increase in solid RM(II)-BTC is minor (110 to 140 Ω). In fact, the values of $R_s$, $R_{int}$ and $R_{ct}$ remain almost constant during the discharging process based on solid RM(II)-BTC and soluble RM. Thus, compared to solid RM(II)-BTC, the larger increasement of $R_t$ is clearly attributable to Warburg impedance ($W$) contributions of soluble RM, which are normally related with carbon dioxide or soluble RM diffusion processes. These results signify that the solid RM(II)-BTC play more important role in accelerating the $CO_2$RR kinetics than this soluble RM at high current densities. To highlight the ability of maintaining rapid reaction kinetics by solid RM(II)-BTC, long-term cyclic stability is explored at a large current of 500 mA g$^{-1}$ with a constant capacity of 1000 mAh g$^{-1}$. As shown in Fig. 3f, the solid RM(II)-BTC cathode can operate stably up to 400 cycles with a discharge terminal potential of 2.70 V and an overall charge plateau below 3.7 V, whereas the pure CNTs cathode can only last for 18 cycles (Supplementary

Fig. 8). As presented in Supplementary Fig. 9, the stability of solid RM(II)-BTC can remain stable after 400 cycles through the XRD test. In conclusion, all experimental results indicate that the solid RM(II)-BTC cathode demonstrates superior catalytic activity for $CO_2$RR/ER. Following this, solid RM(II)-BTC is compared to some commonly reported cathode catalysts of Li-$CO_2$ batteries in terms of various electrochemical properties. It is found that the extremely low discharge/charge potential gap and superb cycling stability have both been integrated into the solid RM(II)-BTC cathode, and its overall performance outperforms that of some noble metal catalysts (e.g., Au, Ru, Ir)[44–48] for Li-$CO_2$ batteries ever reported (Supplementary Fig. 10 and Supplementary Tables 2–4). Moreover, a flexible pouch-type Li-$CO_2$ battery is prepared by using solid RM(II)-BTC. The fabricated pouch battery, as shown in Fig. 3g, h, has a stable electrochemical performance (up to 120 cycles) and also can provide power output for red LED lights even at varied bending angles ranging from 60° to 90° and 180° (Supplementary Fig. 11). The successful preparation of Li-$CO_2$ pouch battery employing solid RM(II)-BTC accelerates the step of flexible metal-$CO_2$ batteries towards practical application.

## Product characterizations of Li-$CO_2$ battery catalyzed by solid RM(II)-BTC

To provide essential insight into the final discharge product in Li-$CO_2$ batteries, the different discharge/charge stages of CNTs and solid RM(II)-BTC cathode are investigated via a series of ex-situ characterizations. Based on Raman, XRD spectroscopy and SEM technique (Supplementary Figs. 12–13), we have observed that the CNTs cathode is gradually deposited by the thin strip-like discharge product $Li_2CO_3$, giving rise to high charge potential and low round-trip efficiency, which is consistent with the previous reported carbon material cathodes[49]. In conjunction with the aforementioned findings, there appeared to be a distinct reaction mechanism or another discharge product that happens in Li-$CO_2$ batteries for the solid RM(II)-BTC cathode, which can provide a lower discharge/charge voltage gap and a more readily decomposed product below 3.7 V.

As depicted in Fig. 4a, compared to ex-situ Raman spectroscopy of the pristine RM(II)-BTC cathode, a small sharp peak indicating $Li_2C_2O_4$ at 1487 cm$^{-1}$ is observed during the discharge process, and the Raman peak fades after recharging, indicating that $Li_2C_2O_4$ may be the primary product[50]. Additionally, noticeable peaks at 605, 1417, 1560 and 1607 cm$^{-1}$, corresponding to $\rho(O - C = O)$, $\delta_s(C - O) + \rho_\omega(C - O)$, and $v_a(O - C = O)$ of $Li_2C_2O_4$ respectively, are detected in the discharged cathode via FT-IR spectroscopy (Fig. 4b)[51,52]. The ex-situ XRD test, as shown in Fig. 4c, also provides irrefutable evidence for the reversible formation and decomposition of crystalline $Li_2C_2O_4$ on solid RM(II)-BTC cathode. Besides several characteristic peaks of solid RM(II)-BTC, three main peaks centering at 19.78°, 27.39°, and 34.76° can be observed, continuously appearing and disappearing as discharge-charge proceeds, assigned to the (011), (−101), and (111) planes of standard $Li_2C_2O_4$ (JCPDS card No. 24-0646)[53,54]. To investigate the morphology of the discharge product, the surfaces of the solid RM(II)-BTC cathode at different stages are analyzed via SEM and TEM characterizations. After discharge to 2.0 V, the smooth surfaces of the pristine RM(II)-BTC cathode are partially covered with tiny particles with an average size of 100 nm (Fig. 4d). The formed particles disappear reversibly after complete recharging, and the glossy surfaces of the cuboid RM(II)-BTC cathode reappear (Fig. 4e and Supplementary Fig. 14). Furthermore, the corresponding selected-area electron diffraction (SAED) pattern in the inset image (Fig. 4d), attributed to the (011), (−101), (111), and (004) planes of polycrystalline $Li_2C_2O_4$. All of the above characterizations elucidate that $Li_2C_2O_4$ is indeed the primary discharge product with assistance of solid RM(II)-BTC and can be decomposed reversibly, which is completely different from the product generated by pure CNTs cathode. Subsequently, the in-situ differential electrochemical mass spectrometry (DEMS) during

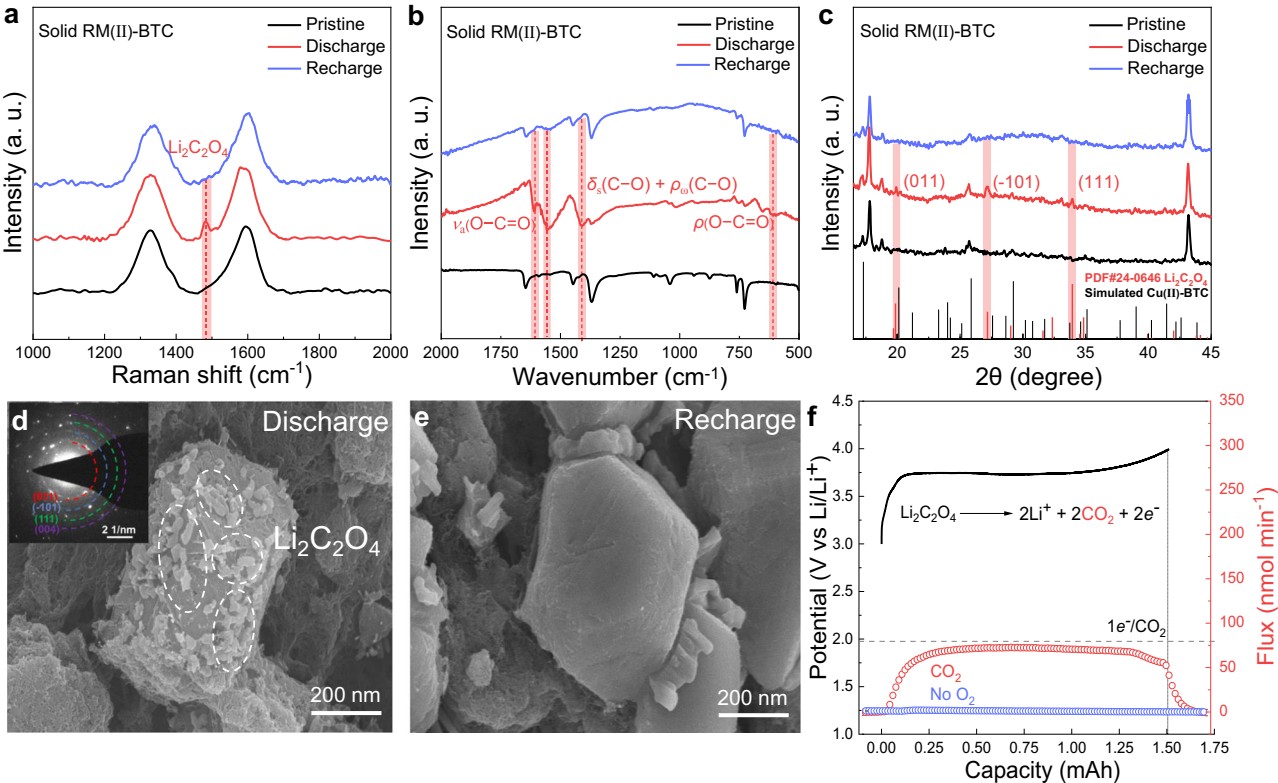

**Fig. 4 | Product characterizations of Li-CO₂ battery using the solid RM(II)-BTC cathode. a** Raman, (**b**) FT-IR and (**c**) XRD spectra of solid RM(II)-BTC cathode at different stages (pristine, discharge, recharge). SEM images of (**d**) discharged and (**e**) recharged of solid RM(II)-BTC cathode, with an inset of corresponding SAED pattern of discharged product. **f** DEMS test during charging of Li-CO₂ battery using solid RM(II)-BTC cathode.

the discharge/charging process is conducted to monitor the gas consumption/evolution and reveal the ratio of transferred electrons of the Li-CO₂ battery with solid RM(II)-BTC cathode. As shown in Supplementary Fig. 15, the consumption of $CO_2$ proves that it is indeed involved in the discharge process. Similarly, no gas is detected other than $CO_2$ during the charging process in Fig. 4f. Moreover, the charging result of DEMS shows that the charge-to-mass ratio is $1.05e^-/CO_2$, which is close to the theoretical value of $1e^-/CO_2$ based on the reversible $CO_2$ evolution reaction (Eq. 4). This further verifies that the primary charging reaction is the decomposition of $Li_2C_2O_4$ product. Finally, we quantify the amount of produced $Li_2C_2O_4$ by through Inductive Coupled Plasma Emission Spectrometer (ICP). When discharging to 1.5 mAh, the actual measured $Li_2C_2O_4$ is 2.75 mg by ICP test, which is close to the theoretical value.

$$Li_2C_2O_4 \rightarrow 2Li^+ + 2CO_2 + 2e^- \qquad (4)$$

## Solid RM(II)-BTC mediated discharge mechanism towards Li₂C₂O₄ route

To uncover the mechanism behind the formation of $Li_2C_2O_4$ on solid RM(II)-BTC cathode, the surface composition of the cathode at different stages of discharge is analyzed using ex-situ XPS, Raman, and UV-vis spectroscopy. First, through the XPS test, we find that the Cu 2p spectra of the pristine RM(II)-BTC cathode in Ar (Fig. 5a, top) is fitted into $2p_{3/2}$ peak at around 934.5 eV, indicating that only one valence state (Cu(II)) exists in pristine RM(II)-BTC cathode. As the discharge deepens (Fig. 5a, middle), a new peak at ~932.9 eV in Ar ascribed to the low valence state of Cu(I) rapidly increases, and the portion of Cu(II) is almost completely replaced by Cu(I). Meanwhile, a new peak at 55.89 eV is identified in the Li 1s spectra when compared to the pristine RM(II)-BTC cathode (Fig. 5b, top and middle). It may be related to the fact that the high

valence state (RM(II)-BTC) has been partially reduced to the low valence state (Li₂RM(I)-BTC)[34]. By comparison, when $CO_2$ is introduced, the proportion of Cu(II) stays the same as the pristine stage (Fig. 5a, bottom), indicating that the reduced Li₂RM(I)-BTC may react with $CO_2$ and then be oxidized to Li₂RM(II)-BTC-2CO₂⁻ immediately. Subsequently, the characteristic peaks of O − C = O at 288.90 eV and Li at 55.34 eV for $Li_2C_2O_4$ are observed in the C 1s and Li 1s spectra[55], respectively (Fig. 5b, c, bottom). It is clearly evidenced that the discharge product is $Li_2C_2O_4$ rather than $Li_2CO_3$. Obviously, the redox of copper ions is critical in the entire process. To further confirm the valence state of Cu in solid RM at various steps cobaltocene ($CoCp_2$) can also be employed. This lavender $CoCp_2$ can rapidly react with the high valence state Cu(II), and form the faint yellow $CoCp_2^+$ via monitoring the solution using UV-visible spectroscopy[56]. As shown in Supplementary Fig. 16a, the solution containing $CoCp_2$ and $CoCp_2^+$ show an absorbance at around 530 and 405 nm, respectively. Upon the addition of discharge product of solid RM(II)-BTC cathode in $CO_2$ to solution containing $CoCp_2$, the characteristic absorption peak of $CoCp_2$ disappears at 530 nm. Instead, a new absorbance peak appears at around 405 nm, which exhibits an oxidation of $CoCp_2$ to $CoCp_2^+$. However, the discharge product of solid RM(II)-BTC cathode in Ar do not cause any color change of $CoCp_2$ solution. This color variation coincides with that observed upon direct electrochemical oxidation of $CoCp_2$ to $CoCp_2^+$ (Supplementary Fig. 16b), implying that the valence state of Cu in solid RM at discharge stage in $CO_2$ atmosphere is still mainly the high valence state Cu(II), while in Ar atmosphere it is dominated by the low valence state Cu(I)[34].

To provide further clarification that $Li_2C_2O_4$ is indeed produced by the interaction of reduced Li₂RM(I)-BTC species and $CO_2$ via low valence state Cu(I), we explored the Raman spectroscopy of solid RM(II)-BTC under different reaction conditions. For the pristine RM(II)-BTC, two pairs of signal peaks at 462/505 cm⁻¹ and 746/828 cm⁻¹ referring to the vibration modes of Cu-O and C-H can be observed,

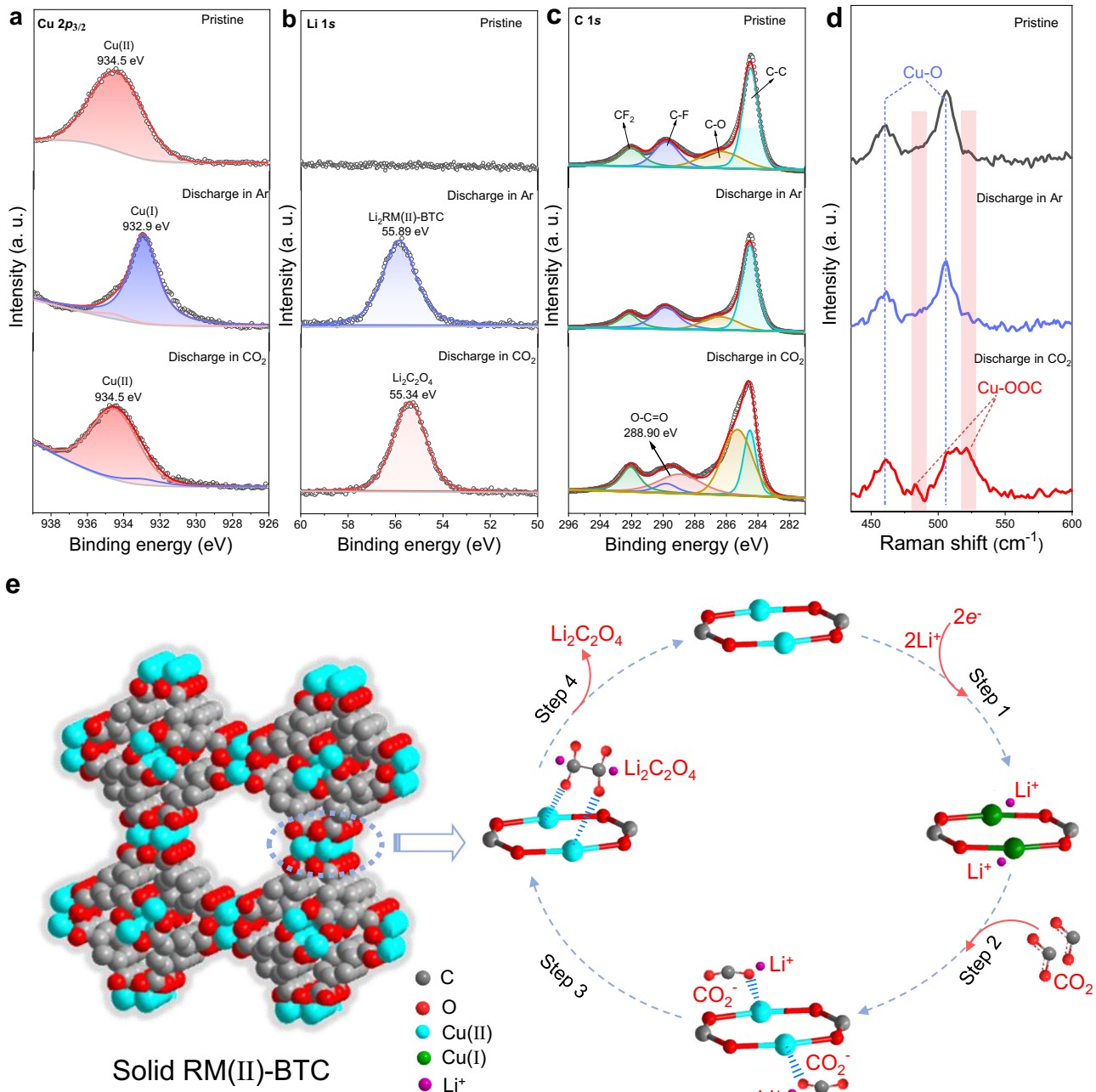

**Fig. 5 | Verification of reaction mechanism and schematic illustration of the reaction pathway.** XPS Cu $2p_{3/2}$ spectra (**a**), Li $1s$ spectra (**b**) and C $1s$ spectra (**c**) of solid RM(II)-BTC cathode at different discharge stages. **d** Raman spectra of solid RM(II)-BTC cathode in the Li-CO$_2$ battery at different stages. **e** The local three-dimensional space-filling diagram of solid RM(II)-BTC cathode and schematic illustration of reaction mechanism to form Li$_2$C$_2$O$_4$ during discharge of solid RM(II)-BTC cathode in the Li-CO$_2$ battery. The gray, red, cyan, green, and purple balls represent C, O, Cu(II), Cu(I) and Li$^+$, respectively.

respectively (Fig. 5d and Supplementary Fig. 17). The position of the Cu-O peak does not change when discharging in Ar, even though, at this point the high valence state Cu(II) has been reduced to the low valence state Cu(I). The discharge process is then continued and CO$_2$ gas is injected. We observe that a couple of new peaks appearing at 482 and 520 cm$^{-1}$ arise in the Raman spectroscopy, which is consistent with that of the bridged Cu-CO$_2^-$ adduct[57]. These results indicate that reduced species Li$_2$RM(I)-BTC capture CO$_2$ to form the [Li$_2$RM(II)-BTC]-2CO$_2^-$ adduct primarily.

Hence, combined with the observed infrared peaks of C$_2$O$_4^{2-}$ and reported literature[23,34], we infer that solid RM(II)-BTC catalyzes the discharge process of Li-CO$_2$ battery, leading to the generation of Li$_2$C$_2$O$_4$ discharge product via a dimerization catalytic reaction, which

is shown in Fig. 5e. The whole discharge reaction mechanism is divided into four steps. Among these steps, step 1 involves an electrochemical reaction, while steps 2, 3, and 4 involve chemical reactions that have negligible reaction time. During the discharge process, the high valence RM(II)-BTC is electrochemically reduced, resulting in the production of a low valence state Li$_2$RM(I)-BTC in step 1. This reduction is achieved through the insertion of lithium ions into the RM(II)-BTC lattice. After step 2, the absorbed CO$_2$ is captured by the reduced Li$_2$RM(I)-BTC to form the bridging unit (Cu(II)-CO$_2^-$) through a chemical reaction. Then, in step 3, two bridging units (Cu(II)-CO$_2^-$) polymerize into a dimeric oxalate intermediate (Cu(II)-C$_2$O$_4^{2-}$-Cu(II)). In step 4, Li ions combine with the dimeric oxalate intermediate, evolving Li$_2$C$_2$O$_4$ product and dissociating from the solid RM(II)-BTC. The

discharge process continues as these cycles (four steps) repeat again, with an increasing discharge capacity.

$$C_2O_4{}^{2-} \rightarrow CO_2{}^{2-} + CO_2 \qquad (5)$$

$$C_2O_4{}^{2-} + CO_2{}^{2-} \rightarrow 2CO_3{}^{2-} + C \qquad (6)$$

In previous work on Li-CO$_2$ batteries, Li$_2$C$_2$O$_4$ was often considered as an intermediate product of the final discharge product Li$_2$CO$_3$. This is because the instability of the C$_2$O$_4{}^{2-}$ on the electrode surface makes it easily undergo disproportionation to form Li$_2$CO$_3$ (Eqs. 5 and 6). So, why can the Li$_2$C$_2$O$_4$ generated by the dimerization catalytic reaction occurring on the surface of solid RM(II)-BTC stably exist as the discharge product of the battery? Obviously, this is because the solid RM(II)-BTC MOF compound stabilizes the Li$_2$C$_2$O$_4$, preventing it from undergoing disproportionation reaction in the organic electrolyte. Here, to compare the capabilities of stabilizing Li$_2$C$_2$O$_4$ on solid RM(II)-BTC MOF and CNTs, the first-principles calculation based on the density functional theory (DFT) is further implemented.

According to the computational results (Supplementary Fig. 18a, b), the adsorption energy of pure CNTs for Li$_2$C$_2$O$_4$ is −0.26 eV. The solid RM(II)-BTC, on the other hand, has a considerably higher adsorption energy for Li$_2$C$_2$O$_4$ of −2.72 eV. This stabilizing effect can also be verified by charge density difference in Supplementary Fig. 19a, b. The outer electrons of solid RM(II)-BTC have transferred into O atoms of the optimized Li$_2$C$_2$O$_4$ structure, thus enabling the rapid stabilization of discharging product Li$_2$C$_2$O$_4$ via the delocalized electrons[18,19]. However, the electron transition between CNTs and Li$_2$C$_2$O$_4$ is not as clear as compared to that between CNTs and Li$_2$CO$_3$ (Supplementary Fig. 19c, d). Likewise, the corresponding adsorption energies of CNTs and solid RM(II)-BTC for Li$_2$CO$_3$ are −1.23 eV and −0.55 eV, respectively (Supplementary Fig. 18c, d), suggesting that the discharge product Li$_2$CO$_3$ forms more readily on CNTs. Therefore, it is considered that the solid RM(II)-BTC plays key role in the stabilization of discharge product Li$_2$C$_2$O$_4$, and avoid further disproportionation reaction to produce insulating Li$_2$CO$_3$, which is agreement with the experimental results.

In conclusion, this study discovered a solid redox mediator, a coordination compound formed by Cu(II) ions and benzene-1,3,5-tricarboxylic acid. It can serve as an efficient catalyst that regulates the discharge and charge reaction pathways of Li-CO$_2$ batteries. With a high specific surface area of 1650 m$^2$ g$^{-1}$, this material possesses more catalytic sites, while its appropriate pore size of 9 Å allows it to adsorb more CO$_2$ molecules. Combined with CNTs, the electrode exhibits a high electronic conductivity of 3 S cm$^{-1}$, which is necessary to reduce polarization. We found that the solid RM(II)-BTC replaces the difficult direct electrochemical reduction of CO$_2$ in Li-CO$_2$ batteries, leading to a significant increase in the output voltage to 2.8 V and the generation of Li$_2$C$_2$O$_4$ that is more easily electrochemically oxidized. This lowers the charging voltage of the battery to 3.7 V. Moreover, solid RM(II)-BTC retain the effectiveness of soluble RMs while avoiding the shuttle consumption issue and slow kinetics since they are anchored to the positive electrode surface. Therefore, Li-CO$_2$ batteries using RM(II)-BTC as a catalyst can discharge/charge stably for over 400 cycles in limited capacity. More importantly, we analyzed and proposed the dimerization catalytic reaction mechanism of RM(II)-BTC during charging and discharging. During the discharge process, RM(II)-BTC receives electrons and is embedded by lithium ions in the solution, forming a Li-containing reduced state Li$_2$RM(I)-BTC. Simultaneously, numerous CO$_2$ molecules adsorbed on the MOF surface undergo a coordination reaction with Li$_2$RM(I)-BTC to form the coordinated structure unit of Cu(II)-CO$_2{}^-$. This unit then experiences a dimerization chemical reaction to produce lithium oxalate and the initial state of RM(II)-BTC, leading to a perfect cyclic process that continuously consumes CO$_2$ to generate Li$_2$C$_2$O$_4$, and provide a discharge specific capacity of up to 9040 mAh g$^-$

[1]. It is known that the Li-CO$_2$ battery using the Li$_2$CO$_3$ path encounters a bottleneck due to the ultra-low discharge voltage and high charge overpotential. Although soluble RM can significantly optimize the discharge and charge voltages by adjusting the Li-CO$_2$ battery to the oxalate path, their dissolution, shuttle drawbacks and slow kinetics caused by concentration polarization are difficult to eliminate fundamentally. The emergence of solid RM not only retains the advantages of soluble ones but also perfectly solves their defects, effectively promoting the progress of Li-CO$_2$ battery technology.

## Methods

### Materials

Copper nitrate (Cu(NO$_3$)$_2$·3H$_2$O), benzene-1,3,5-tricarboxylic acid (BTC) and 2-aminoethanol (NH$_2$-CH$_2$CH$_2$OH) were purchased from Aladdin reagent. A nitric acid-oxidized CNTs aqueous dispersion (2 mg ml$^{-1}$) was purchased from Sigma-Aldrich. The supports were Whatman Anopore AAO membranes with a porosity of 50% and an average pore size of around 200 nm. Throughout the experiments, ultrapure water (18.2 MO) generated by a water purification system (Smart-RO15) was employed. For the components in the electrolytes, LiTFSI and TEGDME were obtained from Sigma Aldrich. The Li metal anode ($\Phi$ 12 mm) was purchased from Aladdin.

### Synthesis of CHNs

CHNs were prepared by rapidly mixing equal volumes of NH$_2$-CH$_2$CH$_2$OH solution (8.54 mg, 1.4 mM) and Cu(NO$_3$)$_2$·3H$_2$O (96.6 mg, 4 mM) solution at 25 °C and ageing for 2 days[30,31].

### Synthesis of solid RM(II)-BTC/CNTs composite material

Initially, a CHN/CNTs composite material was created by filtering a combination of CHN (30 ml) and CNTs (0.5 mg) solutions via an AAO membrane. The prepared CHN/CNTs composite material was then immersed in benzene-1,3,5-tricarboxylic acid water/ethanol solution (10 ml, 5 mM) with a volume ratio of 1:1 for 1 h at 25 °C and finally obtained the solid RM(II)-BTC/CNTs composite[28,29].

### Preparation of electrode material

The electrodes were obtained by rolling the mixture of the solid RM(II)-BTC/CNTs (or pure CNTs) materials with polytetrafluoroethylene aqueous solution (PTFE, 12 wt%) at a mass ratio of 90:10 into a film ($\Phi$ 12 mm, 1.13 cm$^2$), then pressed them on the stainless-steel mesh ($\Phi$ 13 mm, 1.32 cm$^2$) current collector and dried in vacuum oven for 12 h at 120 °C before assemblage. The mass loading of all electrodes is approximately 1.0 ± 0.2 mg cm$^{-2}$ (based RM(II)-BTC/CNTs). The mass loading of all electrodes is approximately 0.287 ± 0.05 mg cm$^{-2}$ (based RM(II)-BTC). For all the controls, the mass loading of all electrodes is approximately 0.3 ± 0.05 mg cm$^{-2}$ (based CNTs). All current densities and capacities were normalized by the mass of active materials on the cathode. The specific energy based on active substance on the cathode was the product of specific capacity and output voltage.

### Electrochemical measurements

All electrochemical experiments are tested at 25 °C. Electrochemical tests were conduced in customized battery mould containing a lithium metal anode ($\Phi$ 12 mm, 1.13 cm$^2$), a pre-obtained cathode ($\Phi$ 12 mm, 1.13 cm$^2$), and electrolyte (1 M LiTFSI in TEGDME impregnated into a glassy fiber separator (Whatman, $\Phi$ 14 mm, 1.54 cm$^2$). The thickness of a glassy fiber separator was 675 μm. All batteries assembly procedures were performed in an argon-filled glove box with a pressure of 1 atm, as well as negligible O$_2$ and H$_2$O levels (< 0.1 ppm). All the batteries should be aerated with the tested gas (CO$_2$ or Ar) for 20 min and then left to rest for 6 h before testing. The cyclic voltammetry (CV) tests were performed on an electrochemical workstation (CHI740E, Chenhua Co., Shanghai, P.R. China) between 2 and 4.5 V at a low rate of 0.1 mV s$^{-1}$ and the iR-compensation were not performed during the CV tests. The

galvanostatic experiments were carried out on the LAND CT2001A Battery Testing Systems (Wuhan LAND electronics Co., Ltd, P.R. China) using the customized battery mould. The EIS Nyquist plots were measured with a frequency range of 100 kHz to 0.1 Hz by an electrochemical station (CHI760E, Chenhua Co., Shanghai, P.R. China).

## Characterization

The discharged and recharged electrodes were washed with DME and dried sufficiently before characterization. XRD analysis was performed on a Bruker D8 advanced diffractometer using Cu-Kα radiation ($\lambda = 1.5406$ Å), with the voltage and current kept at −40 kV and 40 mA, respectively. Raman spectroscopy was carried out on a Renishaw in Via confocal microscope with an air-cooled He-Ne laser at 633 nm excitation. Fourier transform infrared (FTIR) measurements were performed using a FTIR spectroscope (PerkinElmer, Spectrum Two LiTa) in the range of 2000−500 cm$^{-1}$ with a resolution of 1 cm$^{-1}$. XPS were performed on a Thermo Fisher Scientific Model K-Alpha instrument configured with Al Ka radiation (1486.6 eV) and XPS data was corrected with 284.8 eV as the basis. $N_2$ adsorption-desorption and $CO_2$ adsorption data were carried out at 77 K and 298 K under 1 atm, respectively. The morphologies and structures were obtained from a Hitachi SU8010 field emission SEM, in which the accelerating voltage was set at 10.0 kV and the current density of 10 mA. The microstructure was further performed using 200 kV transmission electron microscopy instrument (TEM, FEI TF20). Thermogravimetric analysis was performed in air atmosphere using a TA Instrument SDT Q600, which the temperature ranges from 25 to 1000 °C and the heating rate was 10 °C min$^{-1}$. To evaluate the interaction between the Cu(I) or Cu(II) species and $CO_2$ during the discharge process, ultraviolet-visible (UV-vis) absorption spectrum data were carried out on a UV-vis spectrophotometer (UV-2700, SHAMADZU) within the spectral range of 350−700 nm. The electrochemical mass spectrometry (DEMS) measurements were performed by a quadrupole mass spectrometer (PrismaPro QMG 250 M2) with a turbomolecular pump (Pfeiffer Vacuum). Spectrometer (ICP) were carried out on ICP-OES Avio™ 200 (PerkinElmer). Firstly, discharging to 1 mAh, disassemble the battery, and clean the cathode with dry DME solvent to remove the electrolyte on the surface of cathode. Secondly, the cathode was soaked in dilute hydrochloric acid (20 ml, 1 mol L$^{-1}$) for 24 h, and then and then diluted to 1 L with deionized water. Finally, Inductive Coupled Plasma Emission Spectrometer (ICP) were carried out on ICP-OES Avio™ 200 (PerkinElmer) and the ICP result showed the Li$^+$ concentration is 0.375 mg L$^{-1}$.

## DFT calculation details

We performed all DFT calculations using the Vienna Ab-initio Simulation Package (VASP5.4) package with Projector Augmented Wave (PAW) pseudo-potentials[58,59]. The exchange-correlation energy of the electrons was described by employing the Generalized Gradient Approximations (GGA) using the Perdew-Burke-Ernzerhof (PBE) function[60]. Due to the large size of solid RM(II)-BTC, representative clusters were selected for calculation. A vacuum layer with a thickness of about 15 Å was provided in all three directions to exclude the influence of periodic mirror images and the gamma k-point was selected for our calculations. The Grimme method was applied to take long-range interactions into consideration, and the cutoff kinetic energy was set to 520 eV. The Brillouin zone (BZ) was sampled with a $2 \times 2 \times 1$ k-point for geometry optimizations. The total energies were converged to within 10$^{-6}$ eV, and the force tolerance for geometry optimization is 0.02 eV Å$^{-1}$.

Absorption energies $\Delta E$ were calculated using equation as follows:

$$\Delta E = E_{AB} - E_A - E_B$$

where $E_A$, $E_B$ and $E_{AB}$ represent the total energies of matrix (solid RM(II)-BTC or CNT), adsorbed molecules ($Li_2C_2O_4$ or $Li_2CO_3$), and combined system of matrix and adsorbed molecules, respectively.

The charge density differences $\Delta\rho$ were calculated using equation as follows:

$$\Delta\rho = \rho_{AB} - \rho_A - \rho_B$$

where $\rho_A$, $\rho_B$ and $\rho_{AB}$ represent charge densities of matrix (solid RM(II)-BTC or CNT), adsorbed molecules ($Li_2C_2O_4$ or $Li_2CO_3$), and combined system of matrix and adsorbed molecules, respectively.

## Data availability

The data generated in this study are provided in the paper and Supplementary Information. Additional relevant data are available from the corresponding author on request.

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

## Acknowledgements

This research was supported by the national key R&D program of China (2022YFB2502104), the national natural science foundation of China (22179059, 22239002, 22209070), science and technology innovation fund for emission peak and carbon neutrality of Jiangsu province (BK20231512, BK20220034), key R&D project funded by department of science and technology of Jiangsu Province (BE2020003).

## Author contributions

P.H. conceived the idea and conducted the research. P.H. and H.Z. supervised the projects. W.L. performed the experiments. M.Z. performed the DFT calculations. W.L., P.H., and H.Z. analyzed experiment results. W.L. and P.H. wrote the manuscript. W. L., M.Z., X.S., C.S., L.W., X.M., P.H. and H.Z. discussed the results and commented on the manuscript.

## Competing interests

The authors declare no competing interests.
