## [Peer Review File · Nature Communications]

REVIEWER COMMENTS

Reviewer #1 (Remarks to the Author):

This paper reports a solid redox mediator (RM) for Li-CO₂ batteries that produce Li₂C₂O₄ products rather than Li₂C₂O₃. It has been claimed that the solid redox mediator is more efficient than liquid RMs for this type of batteries. In particular, it was argued that the electrochemically reduced RM(I)-BTC effectively captures CO₂ which promotes the formation of Li₂C₂O₄ through a dimerization reaction. However, there are several serious concerns about the electrochemical, battery and characterization results which need to be addressed by authors. At this stage I cannot recommend the manuscript for publication in Nature Communication. More detailed comments are as below:

1. Figure 1c demonstrates the architecture of the battery set up. It has been mentioned that solid RM(II)-BTC/CNTs was pressed on the stainless-steel current collector. This design of electrode is similar to that of Li-ion batteries. However, in Li-CO₂ batteries the cathode electrode should be porous for gas transport to the electrolyte medium. It is not clear how the design of the battery shown in Figure 1c can mimic a typical Li-CO₂ battery. This needs clarification.

2. The mass loading of the cathode material was mentioned to be ~1mg/cm². Based on this loading, the current density and capacity of reported results in Figure 3f are estimated to be ~ 0.5 mA/cm² and 1mAh/cm², respectively. These are fairly big numbers. However, given the non-porous nature of the cathode electrode, it is not clear how the source of CO₂ is provided for the reactions.

3. The electrolyte is TEGDME and its CO₂ solubility is not high. For such a high capacity and current density reported in this paper, it is expected that very high number of CO₂ molecules are needed for the reactions. However, TEGDME does not have that capacity for CO₂ solubility. This needs further clarification.

4. In Figure 3, the capacity loss of discharge curve for all reported batteries are zero. At a high current density this is unusual and needs further clarification.

5. Comparison of discharge and charge curves in Figure 3b and 3e for both CNT and RM(II)-BTC shows similar results. However, the current density in Figure 3e is 15 times more than Figure 3b. These results do not make sense and seem not to be correct.

6. In Figure 3a, authors need to describe why oxidation peaks in Ar and CO₂ media are different and shows a big shift in CO₂ with respect to Ar. Also, in CO₂ medium, two peaks were assigned for oxidation, however, the first peak for E_{a2} is not visible. This needs clarification.

7. In Figure 3a, the scan rate is very low (0.1mV/s). It is recommended that the authors redo the experiments at 10 mV/s and study the redox peaks again.

8. Authors need to quantify the amount of produced Li₂C₂O₄ at the end of discharge process.

9. Mechanism of Li₂C₂O₄ formation during discharge has not been discussed in the manuscript. It is recommended that authors describe the entire process.

Reviewer #2 (Remarks to the Author):

The manuscript discusses the application of a Cu(II)-MOF as a solid state electrocatalyst for Li-CO₂ batteries. The authors demonstrate, via both electrochemical and structural tools, that the discharge product is predominantly Li₂C₂O₄ instead of the conventional Li₂CO₃. This is encouraging for the field of metal-CO₂ batteries and thus, is an interesting piece of work. There are, however, several serious concerns which lead to great degree of inconclusiveness with the conclusions of the work. The points are mentioned below:

1. The authors seem to have overhyped their criticism on the crossover issue associated with solution based RMs. Crossover depends on many parameters let alone the chemical composition / design of the RMs. Crossover can be also due to CO₂ itself and this will be a natural phenomenon irrespective of whether there is a solid state or a liquid state electrocatalyst/RM. So, the Introduction needs to tone down on the crossover matter and focus more on the problem of the redox process of CO₂ and how to make it reversible.

2. The RM(II)-BTC has been referred to as a solid state to RM. The redox process demonstrated in Fig 3a does not prove that the RM(II)-BTC is a RM. The Li-CO₂ potential is approx. 2.8 V whereas the Cu(II)|Cu(I) is at 2.85 V. This is not a RM as per the standard definition for RMs. The RM(II)-BTC is another example of a solid-state electrocatalyst and hence the reference to RM(II)-BTC as a RM should be withdrawn. The RM(II)-BTC should be referred to as a solid electrocatalyst.

3. In the voltage range depicted in Fig 3a, the authors report the Cu(II)|Cu(I) at 2.85 V. This is a bit surprising as usually one observes the transition from Cu(I)|Cu(0) at such voltage values. In this wide range, one should observe the Cu(II)|Cu(I) as well as Cu(I)|Cu(0). The authors have not shown the derivative plot. From the Fig 3a it seems there is a shallow peak between 3.25 to 3.5 V. This may be indeed the Cu(II)|Cu(I) redox couple and not the one (at 2.85 V) referred in the manuscript. It is strongly

felt that both redox couples are operative in the RM(II)-BTC and the electrocatalysis involves both Cu(II)|Cu(I) as well as Cu(I)|Cu(0).

4. In the manuscript the discharge peak at 3.63 V has been referred to as due to the discharge products. This is very vague and speculative. The authors should demonstrate evidence to backup this statement.

5. The statement "An inconspicuous slope appears at the start of the charging curve in the battery containing solid RM(II)-BTC, possibly due to the oxidation of Cu(I) to Cu(II) after deep discharge" is very vague. Why should this be due to a redox couple and not due to existence of multiphases? What is the evidence ?

6. Fig 3b: The CNT discharge potential is too low and is usually expected at 2.3 to 2.5 V. This is a very surprising observation.

7. The authors have demonstrated the Li₂RM(I)-BTC reaction with CO₂ and its oxidization to Li₂RM(II)-BTC-2CO₂- using a chemical experiment using CoCp. However, this may not exactly replicate the electrochemical reaction. The authors should demonstrate an in situ UV-VIS spectroscopy data to substantiate their claims.

8. In Fig 5a, the authors show only the Cu XPS data. What are the changes in the Li1s and C1s data. These changes should be shown in the manuscript for a precise understanding of the reaction pathways and energy storage characteristics.

9. The dimer model is primarily shown using DFT studies. There are practically no experimental support for this proposition remains inconclusive. There are several questions here. What is the evidence of LiCO₂- being anchored to the RM(II)-BTC and that the eventual formation of Li₂C₂O₄ being anchored to it. This can also be in the solution very close to the porous electrode.

10. Supplementary Information Figure 4 mentions in situ Raman however the data shows XRD. The error needs to be rectified.

11. What is the basis for the selection of the EIS circuit shown in Supplementary Information Figure 6 ? It does not seem appropriate. What are the error bars and what are the capacitance values ?

12. Generally the manuscript figures are poor quality. Figure 1a is more a figure suited for a review paper and is not appropriate here. This should be removed from Figure 1. Many of the other panels should be reduced. Additionally, Figure 1 should be reorganized for better clarity. Similarly, Figure 3g is redundant as the authors put up a comparison Table in the ESI. This should also be removed from Fig 3.

The manuscript in the present form is not acceptable for publications. The authors are required to perform additional experimentation to make their claims on a stronger footing.

Reviewer #3 (Remarks to the Author):

The study of the electrochemical reduction of carbon dioxide in organic electrolytes has not only significant scientific importance but also practical significance for achieving carbon peak and carbon neutrality goals. I believe that this article makes an important contribution to the field of lithium-carbon dioxide batteries. In the past, the redox mediators used were soluble materials, and these soluble redox mediators improved the energy storage efficiency of the batteries by altering the reduction and deposition pathways of carbon dioxide, reducing the electrochemical overpotential. The proposed solid-state redox mediator in this paper presents a promising approach. It not only avoids the shuttle effect associated with liquid redox mediators but also enhances the reaction rate. The evidence presented in the paper supports the proposed reaction pathway where carbon dioxide molecules, assisted by copper complexes, form oxalate ions. I believe that this innovative approach will attract widespread attention from researchers in the field of metal-air batteries and open a new realm of solid-state redox mediator design research for lithium-carbon dioxide batteries and even metal-air batteries. Therefore, I recommend it for publication in Nature Communications after the following revisions.

1. The sentence “the charging result of DEMS shows that the charge-to-mass ratio is $0.95e^-/CO_2$, which is close to the theoretical value of $1e^-/CO_2$ based on the reversible CO_2 evolution reaction (Equation 4)” The authors should explain this point and provide the calculation process of the charge-to-mass ratio ($0.95e^-/O_2$) in greater detail.
2. To prove that the CO_2 is indeed involved in the reaction, please supply the DEMS data of discharging process.
3. What is the mass loading of the active material of the flexible pouch-type Li- CO_2 battery?
4. Aside from the electrochemical stability, the stability of solid $RM(II)-BTC$ after cycles are also essential to Li- CO_2 batteries, therefore, the related stability is required to be supplemented.
5. Some minor mistakes and typos:
 - 1) The sentence “These results indicate that reduced species $Li_2RM(I)-BTC$ capture CO_2 to form the $[Li_2RM(II)-BTC]-2CO_2$ - adduct primarily”. The $[Li_2RM(II)-BTC]-2CO_2$ - adduct should be written to $[Li_2RM(II)-BTC]-2CO_2$.
 - 2) The format of physical units, such as $S\ cm^{-1}$, S/cm and $mAh\ g^{-1}$ in this manuscript, needs to be unified.

Responses to Reviewers' Comments

We would like to thank all reviewers for their constructive comments and suggestions, which has in our view significantly raised the quality of the manuscript (NCOMMS-23-24308-T). We have modified the manuscript accordingly, and listed the detailed corrections below point by point for each reviewer. Moreover, all revised portion has been marked in yellow in the revised manuscript. The main corrections and the responses to the reviewers' comments are as follows.

Point-to-Point Responses (shown in the next page)

Reviewer #1 (Remarks to the Author):

This paper reports a solid redox mediator (RM) for Li-CO₂ batteries that produce Li₂C₂O₄ products rather than Li₂C₂O₃. It has been claimed that the solid redox mediator is more efficient than liquid RMs for this type of batteries. In particular, it was argued that the electrochemically reduced RM(I)-BTC effectively captures CO₂, which promotes the formation of Li₂C₂O₄ through a dimerization reaction. However, there are several serious concerns about the electrochemical, battery and characterization results which need to be addressed by authors. At this stage I cannot recommend the manuscript for publication in Nature Communication.

Response: We feel great thanks for your professional review on our manuscript. As you are concerned, there are several problems that need to be addressed. According to your nice suggestions, we have made extensive corrections to our previous manuscript, the detailed point-to-point responses are listed below.

1. Figure 1c demonstrates that architecture of the battery set up. It has been mentioned that solid RM(II)-BTC/CNTs was pressed on the stainless-steel current collector. This design of electrode is similar to that of Li-ion batteries. However, in Li-CO₂ batteries the cathode electrode should be porous for gas transport to the electrolyte medium. It is not clear how the design of the battery shown in Figure 1c can mimic typical Li-CO₂ batteries. This needs clarification.

Response: We sincerely thank the reviewer for careful review. We are really sorry for the confusion caused to you by our mistake. In fact, during the preparation of the cathode electrode, **the solid RM(II)-BTC cathode was pressed on the stainless-steel mesh**. Gases can pass through stainless steel mesh, and in addition, porous electrode active materials are also permeable to gases. As shown in Fig. R1, this porous stainless-steel mesh can achieve the gas transport to the electrolyte medium and cathode electrode surface. As reminded by the reviewer, we have **corrected the ‘stainless-steel current collector’ into ‘stainless-steel mesh current collector’** in the revised

manuscript.

At the same time, the schematic of Li-CO₂ battery with solid RM(II)-BTC cathode in origin Fig. 1c did not exhibit its real structure. Therefore, we have redraw and corrected the schematic of Li-CO₂ battery in revised manuscript, as shown in Fig. R2 (Fig. 1a, b).

Fig. R1. Schematic of RM(II)-BTC cathode with porous stainless-steel mesh collector.

Fig. R2. (a) Schematic of Li-CO₂ battery with solid RM(II)-BTC cathode. RM(II)-BTC can inhibit the shuttle effect since it is fixed at the cathode as a solid state and exhibit high CO₂ adsorption, fast kinetics and reversible redox couples. (b) Solid RM(II)-BTC-mediated reaction mechanism towards Li₂C₂O₄ route. RM(II)-BTC is electrochemically reduced to RM(I)-BTC, which immediately reacts with CO₂ to form RM(II)-BTC-C₂O₄²⁻. Finally, discharge product Li₂C₂O₄ and origin RM(II)-BTC are obtained. The frame with dash-dotted line reflect that after receiving electrons and incorporating lithium ions, RM(II)-BTC can transfer to a compound of lithium and RM(I)-BTC. The gray, red, cyan, green, and blue balls represent C, O, Cu(II), Cu(I) and Li⁺, respectively.

2. The mass loading of the cathode material was mentioned to be ~1mg/cm². Based on this loading, the current density and capacity of reported results in Figure 3f are estimated to be ~0.5 mA/cm² and 1mAh/cm², respectively. These are fairly big numbers.

However, given the non-porous nature of the cathode electrode, it is not clear how the source of CO₂ is provided for the reactions.

Response: We thank the reviewer for this careful review. As we mentioned in the last question, we used porous stainless-steel mesh as collector to achieve the gas transport to the electrolyte medium and the cathode electrode surface. CO₂ gas can undergo electrochemical reduction at the interface between the electrolyte and the electrode, releasing capacity. While CO₂ is continuously consumed at the electrode-electrolyte interface, there is also a continuous influx of CO₂ from outside the battery through the cathode.

3.The electrolyte is TEGDME and its CO₂ solubility is not high. For such a high capacity and current density reported in this paper, it is expected that very high number of CO₂ molecules are needed for the reactions. However, TEGDME does not have that capacity for CO₂ solubility. This needs further clarification.

Response: We thank the reviewer for this comment very much. For the reviewer's comment, we can answer from the following three aspects.

The reaction gas (CO₂) diffuses to the electrolyte and the surface of the porous electrode during the discharge process, and the reduction reaction occurs at the gas-liquid-solid three-phase interface. As the discharge reaction proceeds, the CO₂ dissolved in the electrolyte is consumed. Meanwhile, external CO₂ will continuously dissolve into the electrolyte. Furthermore, the solid RM(II)-BTC cathode has large specific surface area, abundant catalytic sites and suitable pore size, which can adsorb and store more CO₂ molecules. Most importantly, the solubility of CO₂ in TEGDME solvent is much higher than that of O₂ in TEGDME solvent. Many examples of high current and high capacity have been reported in Li-O₂ batteries and Li-CO₂ batteries¹⁻⁴ (*Nature materials*, 2013, 12, 1050; *Nature Energy*, 2017, 2, 17118; *Energy Environment Science*, 2018, 11, 3231; *Energy Environment Science*, 2019, 12, 1046).

4. In Figure 3, the capacity loss of discharge curve for all reported batteries are zero

at a high current density, this is unusual and needs further clarification.

Response: We thank the reviewer for this valuable comment. In order to exhibit the effect of RM(II)-BTC cathode on reducing overpotential in battery, galvanostatic discharge/charge experiments were conducted at a fixed capacity of 1000 mAh g^{-1} using different currents. As we have observed, there is no specific capacity loss but voltage polarization at high current in Fig. R3a-c (Fig. 3c-f). We would be happy to explain the reasons behind this phenomenon to the reviewers.

This capacity-limited (fixed capacity of 1000 mAh g^{-1}) charge-discharge test is primarily used to compare the polarization of electrode reactions at different current densities. In other words, among different sets of test batteries using different current densities, the capacity remains constant, which is why reviewer observed no capacity decay at high currents. However, the charge-discharge voltage of these batteries varies with different currents. The magnitude of this variation primarily results from the catalytic activity of the catalyst material within the electrode. This is indeed an effective method for studying the electrochemical reaction kinetics at the electrode-electrolyte interface for catalytic reactions.

Fig. R3 (a) Comparison of the galvanostatic discharge/charge profiles of CNTs and solid RM(II)-BTC cathode with a fixed capacity of 1000 mAh g^{-1} at 100 mA g^{-1} . (b) Comparison of the galvanostatic discharge/charge profiles of solid RM(II)-BTC at various current densities. (c)

Comparison of the galvanostatic discharge/charge profiles of CNTs (without RM), soluble RM and solid RM(II)-BTC at high current densities (1000 and 1500 mA g⁻¹).

5. Comparison of discharge and charge curves in Figure 3b and 3e for both CNT and RM(II)-BTC shows similar results. However, the current density in Figure 3e is 15 times more than Figure 3b. These results do not make sense and seem not to be correct.

Response: We thank the reviewer for this constructive suggestion. Please allow me to provide an explanation for this experimental phenomenon.

For the CNT electrode, there are still noticeable differences in the charge-discharge curves at current densities of 100 mA g⁻¹ and 1500 mA g⁻¹. The reason why the reviewer perceived the charge-discharge curves of the CNT electrode in Figure 3b and Figure 3e as similar is due to a 10-fold difference in their horizontal axis scales. This difference in axis scales is primarily because Figure 3b presents the full discharge-charge curve controlled by the cutoff voltage (larger specific capacity), while Figure 3e presents the discharge-charge curve with capacity cutoff of 1000 mAh g⁻¹. If you carefully examine the discharge behavior within the same 1000 mAh g⁻¹ specific capacity range (blue dashed line in Fig. R4a and Fig. R4b), you'll notice that the CNT electrode, at a current density of 100 mA g⁻¹, exhibits discharge voltages between 1.4V and 1.2V. Whereas at a current density of 1500 mA g⁻¹, the discharge voltages range from 0.9V to 0.6V.

Fig. R4 (a) Full deep discharge/charge profiles of CNTs and RM(II)-BTC under CO₂ at a current density of 100 mA g⁻¹. (b) The galvanostatic discharge/charge profiles of CNTs and RM(II)-BTC at a current density of 1500 mA g⁻¹ with a fixed capacity of 1000 mAh g⁻¹.

To make these differences clearer to the reviewers, we have **plotted the discharge-**

charge curves with fixed capacity of 1000 mAh g^{-1} of the CNT electrode at both 100 mA g^{-1} and 1500 mA g^{-1} current densities on the same Figure (as shown in Fig. R5a). From this graph, it can be observed that at both current densities, the CNT electrode has voltage differences of 550 mV during discharge and 600 mV during charge. Similarly, in Fig. R5b, for the solid RM(II)-BTC cathode, at two different current densities, the corresponding voltage differences are 250 mV and 450 mV . These also indicate that the RM(II)-BTC electrode performs better in suppressing polarization at higher currents.

Fig. R5 Comparison of the galvanostatic discharge/charge profiles of CNTs (a) and solid RM(II)-BTC (b) at 100 mA g^{-1} and 1500 mA g^{-1} with a fixed capacity of 1000 mA g^{-1} .

6. In Figure 3a, authors need to describe why oxidation peaks in Ar and CO_2 media are different and shows a big shift in CO_2 with respect to Ar. Also, in CO_2 medium, two peaks were assigned for oxidation, however, the first peak for Ea2 is not visible. This needs clarification.

Response: We thank the reviewer for this valuable suggestion very much. It's possible that we didn't explain these cyclic voltammetry curves clearly enough.

So, we have repeated this CV experiment, which may provide clearer information. Here we will provide a more detailed explanation. In Fig. R6 (Fig. 3a), the cyclic voltammetry curve of the solid RM(II)-BTC cathode in a CO_2 atmosphere (red curve) exhibits a reduction peak E_{c2} (2.81 V), corresponding to the electrochemical reduction process of RM(II)-BTC to $\text{Li}_2\text{RM(I)-BTC}$, which is the reaction equation R1 (equation 3 in manuscript) described in the paper. It's worth noting that, at this point, the reduction

product $\text{Li}_2\text{RM(I)-BTC}$ will be oxidized by the abundant CO_2 gas at the electrode interface, reverting back to RM(II)-BTC (as described in steps 2 and 3 in Fig. 5e). However, this oxidation process is a chemical reaction and not an electrochemical process involving electron transfer on circuit, so it doesn't appear as a corresponding peak on the cyclic voltammetry curve. At this point, along with the electrode components reverting to RM(II)-BTC , there may still be a small amount of unoxidized residual $\text{Li}_2\text{RM(I)-BTC}$ and the reduction product of CO_2 , lithium oxalate ($\text{Li}_2\text{C}_2\text{O}_4$). Looking at the anodic part of this cyclic voltammetry curve, you will notice an oxidation peak with somewhat poor symmetry. This oxidation peak is somewhat broad, and its left shoulder is somewhat prominent. This is **a typical double-peak overlap phenomenon (E_{a2} and E_{a2}')**. We can infer that the oxidation peak E_{a2} (3.40 V) corresponds to the electrochemical oxidation and lithium removal reaction of $\text{Li}_2\text{RM(I)-BTC}$, which is the reverse reaction of equation R1 (equation 3 in revised manuscript). This is similar to the oxidation reaction of the electrode in an Ar gas environment (E_{a1} peak in the green curve, 3.34 V). However, the oxidation peaks in Ar and CO_2 gas are different and shows a big shift in CO_2 with respect to Ar. This is owing to the presence of **CO_2 discharge products ($\text{Li}_2\text{C}_2\text{O}_4$) on the electrode surface with low electrical conductivity, resulting in enhanced electrode polarization.**

The oxidation peak E_{a2}' (3.70 V) corresponds to the electrochemical oxidation reaction equation R2 (equation 3 in revised manuscript) of the reduction product of CO_2 , $\text{Li}_2\text{C}_2\text{O}_4$.

Fig. R6 The CV plots of CNTs and solid RM(II)-BTC cathode in Ar and CO₂ atmosphere at 0.1 mV s⁻¹.

In an Ar gas atmosphere, the pair of reduction and oxidation peaks for RM(II)-BTC can be observed more clearly (2.81 V and 3.34 V). However, since there is no CO₂ present here, there will be no formation of Li₂C₂O₄ during the cathodic scan process. Therefore, there won't be a second oxidation peak corresponding to the electrochemical decomposition of lithium oxalate during the anodic scan process either.

In the absence of RM and with only a CNT electrode, the electrode exhibits a pair of reduction and oxidation peaks (2.3V and 4.5 V). The reduction peak corresponds to the reduction of CO₂ to form lithium carbonate and carbon, while the oxidation peak corresponds to the anodic decomposition of lithium carbonate. It is in accordance with previous studies^{5,6}.

The update CV data mentioned above, along with the corresponding explanations and interpretations, have been incorporated into the revised version of the paper (line 16, page 7).

7. In Figure 3a, the scan rate is very low (0.1mV/s). It is recommended that the authors redo the experiments at 10 mV/s and study the redox peaks again.

Response: We thank the reviewer for this valuable suggestion very much. We would like to provide an explanation regarding the choice of cyclic voltammetry scan rates. The organic electrolyte CO₂ electrochemical reduction and evolution reactions investigated in this work involve a multitude of complex multiphase reactions. These

include the generation of solid reduction products at the gas-solid catalyst interface, the insertion and extraction processes of Li ions in the bulk solid redox media, and the decomposition of solid products into gases at the electrode interface. In contrast, traditional aqueous CO₂ reduction to organic compounds typically occurs at the catalyst/water solution interface, falls into the realm of surface reactions, and does not involve solid-phase reactions. This implies that the kinetics of the non-aqueous multiphase reactions studied in this work are slower and more sluggish. To distinguish and study a series of reaction steps through cyclic voltammetry experiments, it often requires the use of slower scan rates. This is why this paper employs a scan rate of 0.1 mV/s. In previous literatures on Li-CO₂ battery, such scan rates have also frequently been used for electrochemical mechanism studies⁷⁻⁹ (*Advanced Materials*, 2022, 34, 2204134; *Advanced Materials*, 2019, 31, 1805484; *Advanced Functional Materials* 2017, 27, 1700564).

However, based on the reviewers' suggestions, we still attempted cyclic voltammetry tests on the RM(II)-BTC electrode in a CO₂ atmosphere at a scan rate of 10 mV/s. The CV spectrum is shown in the Fig. R7. From the Fig. R7, you can observe a pair of reduction and oxidation peaks (2.43 V and 3.65 V). The reduction peak corresponds to the electrochemical reduction of the solid redox media RM(II)-BTC, transforming into Li₂RM(I)-BTC. The oxidation peak corresponds to the electrochemical oxidation of the CO₂ reduction product, lithium oxalate. It is worth noting that at the higher scan rate, the shoulder-to-shoulder double oxidation peaks, which were visible at a scan rate of 0.1 mV/s, are no longer apparent. We believe that the kinetics of the lithium removal oxidation reaction of Li₂RM(I)-BTC are sluggish, resulting in excessive polarization, and this increases the overlap between its oxidation peak and the electrochemical oxidation peak of lithium oxalate.

Fig. R7 The CV plots of solid RM(II)-BTC cathode in CO₂ atmosphere at 10 mV s⁻¹.

8. Authors need to quantify the amount of produced Li₂C₂O₄ at the end of discharge process.

Response: We thank the reviewer for this valuable suggestion very much. According to your suggestion, we carried out an elemental analysis test on the discharge product through Inductive Coupled Plasma Emission Spectrometer (ICP). When discharging to 1.5 mAh, **the theoretical discharged product Li₂C₂O₄ is 2.85 mg**. Through the ICP test, **the actual measured Li₂C₂O₄ is 2.75 mg**, which is based on the measured mass of lithium ions (0.375 mg). A certain quantity of Li₂C₂O₄ may be lost during the test due to the necessity to clean the discharge electrode.

Meanwhile, we have added the related description (page14) and ICP test method (page 21) in revised manuscript as follows.

“Finally, we quantify the amount of produced Li₂C₂O₄ by through Inductive Coupled Plasma Emission Spectrometer (ICP). When discharging to 1.5 mAh, the actual measured Li₂C₂O₄ is 2.75 mg by ICP test, which is close to the theoretical value.”

And

“Firstly, discharging to 1mAh, disassemble the battery, and clean the cathode with dry DME solvent to remove the electrolyte on the surface of cathode. Secondly, the cathode was soaked in dilute hydrochloric acid (20ml, 1mol L⁻¹) for 24h, and then diluted to 1L with deionized water. Finally, Inductive Coupled Plasma Emission Spectrometer (ICP) were carried out on ICP-OES Avio™ 200 (PerkinElmer) and the ICP result showed the

Li⁺ concentration is 0.375 mg L⁻¹.”

9. Mechanism of Li₂C₂O₄ formation during discharge has not been discussed in the manuscript. It is recommended that authors describe the entire process.

Response: We thank the reviewer for this valuable suggestion very much. To understand the reaction pathways exactly, we have **supplemented the Li 1s and C 1s data during discharge** in Fig. R8a-d (Fig. 5a-d). Next, **the formation mechanism of Li₂C₂O₄ during discharge** will be discussed in detail.

First, through the XPS test, we find that the Cu 2p spectra of the pristine RM(II)-BTC cathode in Ar (Fig. R8a, top) is fitted into 2p_{3/2} peak at around 934.5 eV, indicating that only one valence state (Cu(II)) exists in pristine RM(II)-BTC cathode. As the discharge deepens (Fig. R8a, middle), a new peak at ~932.9 eV in Ar ascribed to the low valence state of Cu(I) rapidly increases, and the portion of Cu(II) is almost completely replaced by Cu(I). Meanwhile, a new peak at 55.89 eV is identified in the Li 1s spectra when compared to the pristine RM(II)-BTC cathode (Fig. R8b, top and middle). It may be related to the fact that the high valence state (RM(II)-BTC) has been partially reduced to the low valence state (Li₂RM(I)-BTC)¹⁰. By comparison, when CO₂ is introduced, the proportion of Cu(II) stays the same as the pristine stage (Fig. R8a, bottom), indicating that the reduced Li₂RM(I)-BTC may react with CO₂ and then be oxidized to Li₂RM(II)-BTC-2CO₂⁻ immediately. Subsequently, the characteristic peaks of O-C=O at 288.90 eV and Li at 55.34 eV for Li₂C₂O₄ are observed in the C 1s and Li 1s spectra, respectively¹¹ (Fig. R8c).

Fig. R8 XPS (a) Cu 2p_{3/2} spectra, (b) Li 1s spectra and (c) C 1s spectra of solid RM(II)-BTC cathode at different discharge stages. (d) Raman spectra of solid RM(II)-BTC cathode in the Li-CO₂ battery at different discharge stages.

Fig. R9. (a) Raman, (b) FT-IR and (c) XRD spectrums of solid RM(II)-BTC cathode at different stages (pristine, discharge, recharge).

To provide further clarification that Li₂C₂O₄ is indeed produced by the interaction of reduced Li₂RM(I)-BTC species and CO₂ via low valence state Cu(I), we explored the Raman spectroscopy of solid RM(II)-BTC under different reaction conditions. For the pristine RM(II)-BTC, one pair of signal peak at 462/505 cm⁻¹ referring to the vibration modes of Cu-O can be observed, respectively (Fig. R8d). The position of the Cu-O peak does not change when discharging in Ar, even though, at this point the high valence state Cu(II) has been reduced to the low valence state Cu(I). The discharge process is then continued and CO₂ gas is injected. We observe that a couple of new peaks appearing at 482 and 520 cm⁻¹ arise in the Raman spectroscopy, which is consistent with that of the bridged Cu-CO₂⁻ adduct¹². These results indicate that reduced species Li₂RM(I)-BTC capture CO₂ to form the [Li₂RM(II)-BTC]-2CO₂⁻ adduct primarily.

Based on Raman spectroscopy, XRD spectroscopy and FT-IR techniques (Fig. R9), the discharge product of $\text{Li}_2\text{C}_2\text{O}_4$ on the surface of RM(II)-BTC cathode has been confirmed.

The discussion regarding the confirmation of lithium oxalate products can be found on page 13 in the revised manuscript.

Fig. R10 The local three-dimensional space-filling diagram of solid RM(II)-BTC cathode and schematic illustration of reaction mechanism to form $\text{Li}_2\text{C}_2\text{O}_4$ during discharge of solid RM(II)-BTC cathode in the Li- CO_2 battery. The gray, red, cyan, green, and blue balls represent C, O, Cu(II), Cu(I) and Li^+ , respectively.

Hence, combined with the observed $\text{Li}_2\text{C}_2\text{O}_4$ product, we infer that solid RM(II)-BTC catalyzes the discharge process of Li- CO_2 battery, leading to the generation of $\text{Li}_2\text{C}_2\text{O}_4$ discharge product via a dimerization catalytic reaction, which is shown in **Fig. R10** (**Fig. 5e** in revised manuscript). The whole discharge reaction mechanism is divided into four steps. Among these steps, step 1 involves an electrochemical reaction, while steps 2, 3, and 4 involve chemical reactions that have negligible reaction time. During the discharge process, the high valence RM(II)-BTC is electrochemically reduced, resulting in the production of a low valence state $\text{Li}_2\text{RM(I)-BTC}$ in step 1. This reduction is achieved through the insertion of lithium ions into the RM(II)-BTC lattice. After step 2, the absorbed CO_2 is captured by the reduced $\text{Li}_2\text{RM(I)-BTC}$ to form the bridging unit (Cu(II)-CO_2^-) through a chemical reaction. Then, in step 3, two bridging units (Cu(II)-CO_2^-) polymerize into a dimeric oxalate intermediate ($\text{Cu(II)-C}_2\text{O}_4^{2-}\text{-Cu(II)}$). In step 4, Li ions combine with the dimeric oxalate intermediate,

evolving $\text{Li}_2\text{C}_2\text{O}_4$ product and dissociating from the solid RM(II)-BTC. The discharge process continues as these cycles (four steps) repeat again, with an increasing discharge capacity. **A detailed description of the catalytic cyclic reaction process of RM(II)-BTC can also be found on page 16 in the revised manuscript.**

Reviewer #2 (Remarks to the Author):

The manuscript discusses the application of a Cu(II)-MOF as a solid state electrocatalyst for Li-CO₂ batteries. The authors demonstrate, via both electrochemical and structural tools, that the discharge product is predominantly Li₂C₂O₄ instead of the conventional Li₂CO₃. This is encouraging for the field of metal-CO₂ batteries and thus, is an interesting piece of work. There are, however, several serious concerns which lead to great degree of inconclusiveness with the conclusions of the work. The points are mentioned below:

Response: We feel great thanks for the reviewer's positive comment on the novelty and significance of our work. As you are concerned, there are several serious concerns which lead to great degree of inconclusiveness with the conclusions of the work. According to your nice suggestions, we have made extensive revisions to our previous manuscript, and the detailed point-to-point responses are listed below.

1. The authors seem to have overhyped their criticism on the crossover issue associated with solution based RMs. Crossover depends on many parameters let alone the chemical composition/design of the RMs. Crossover can be also due to CO₂ itself and this will be a natural phenomenon irrespective of whether there is a solid state or a liquid state electrocatalyst/RM. So, the Introduction needs to tone down on the crossover matter and focus more on the problem of the redox process of CO₂ and how to make it reversible.

Response: We thank the reviewer for this valuable suggestion very much. In fact, some reported studies have found that the moderate diffusion of CO₂ is beneficial to Li anode stability, which can create a stable SEI layer¹³ (*Energy Storage Materials*, 2020, 26, 443-447). Therefore, the crossover problem of Li-CO₂ batteries generally refers to the diffusion of soluble RM⁺ between electrodes, which can lead to adverse side effects, such as soluble RM decomposition and lithium metal deterioration. To circumvent this issue, Li-deficient Li_xFePO₄ (x=0.9) as the anode has been employed in previously reported researches¹⁴⁻¹⁵ (*Nature Communications*, 2023, 14, 536; *Nature Materials*,

2016, 15, 882-888.). However, this is not a true Li metal-CO₂ battery. Therefore, solid RM that is stationary on cathode has more advantage than soluble RM in suppressing crossover problem.

We greatly appreciate the reviewer's feedback. Shuttle issue is not the sole problem faced by soluble redox mediators; it is just one of the many issues we need to address. We had overstated the crossover issue of soluble RM. According to reviewer's suggestion, we have reduced the description of crossover matter in the introduction and focused more on the redox process of CO₂ and how to make it reversible. We have **deleted these descriptions of crossover matter:**

“This requires additional treatments, such as the incorporation of lithium protective layers or modified separators, to prevent reactions with the lithium-metal anode, resulting in an increased battery weight and resistance.”

And

“This solid RM(II)-BTC exhibits the similar efficacy as traditional soluble RMs, but its solid phase prevents the shuttle effect and consumption on the negative electrode.”

In the introduction of revised manuscript, besides shuttle issue we also focus on the poor kinetics of soluble RM which is reaction bottleneck of the diffusion process under large current. So, we focused more on the redox process of CO₂ and how to make it reversible. **We have added these descriptions as follows in introduction of revised manuscript:**

“Although certain some soluble RMs enhance electrochemical properties such as round-trip efficiency, discharge capacity, and so on, Li-CO₂ batteries based on these RMs always exhibit unsatisfactory reversibility (often fewer than 70 cycles)¹⁶⁻¹⁸.”

And

“Meanwhile, due to the existence of concentration polarization of soluble RM, it is difficult to tackle the problem of sluggish kinetics at large current. This makes soluble RM unable to diffuse to the cathode surface in time at large current, thus affecting the reduction reaction of CO₂ and the reversibility of battery, leading to the application of soluble RM in Li metal-CO₂ batteries is not practical^{19, 20}.”

2. The RM(II)-BTC has been referred to as a solid state to RM. The redox process demonstrated in Fig 3a does not prove that the RM(II)-BTC is a RM. The Li-CO₂ potential is approx. 2.8 V whereas the Cu(II)|Cu(I) is at 2.85 V. This is not a RM as per the standard definition for RMs. The RM(II)-BTC is another example of a solid-state electrocatalyst and hence the reference to RM(II)-BTC as a RM should be withdrawn. The RM(II)-BTC should be referred to as a solid electrocatalyst.

Response: We thank the reviewer for this valuable suggestion very much. The redox mediator refers to the substance involved in electron transfer in redox reaction. This substance, instead of the battery's active material, gains and loses electrons at the electrode interface (i.e., undergoes electrochemical reactions), and the electrochemical products obtained then react chemically with the battery's active material. This substance also known as electron acceptor or electron donor.

In our manuscript, during the discharge process, the high valence RM(II)-BTC is electrochemically reduced, generating the production of a low valence state Li₂RM(I)-BTC. Meanwhile, the reduction product Li₂RM(I)-BTC will be oxidized by the abundant CO₂ gas (battery's active materials) at the electrode interface, reverting back to RM(II)-BTC. It is clear that RM(II)-BTC itself experiences gains and losses of electrons throughout the discharge process, which participates in the reduction reaction of CO₂.

On the other hand, for the conventional Li₂CO₃ path in Li-CO₂ batteries, the thermodynamic equilibrium potential is calculated to be about 2.80 V (Equation R3). However, the discharge product is Li₂C₂O₄ instead of the conventional Li₂CO₃ in our manuscript, **the thermodynamic equilibrium potential involving Li₂C₂O₄ is calculated to be about 3.04 V** (Equation R4). The redox potential of Cu(II)|Cu(I) is 2.85 V, which is lower than the Li-CO₂ potential based on Li₂C₂O₄ products (3.04 V). Thus, the Li₂RM(I)-BTC can chemically reduce CO₂ to Li₂C₂O₄. Therefore, the RM(II)-BTC can be actually defined as a solid RM.

3. In the voltage range depicted in Fig 3a, the authors report the Cu(II)|Cu(I) at 2.85 V. This is a bit surprising as usually one observes the transition from Cu(I)|Cu(0) at such voltage values. In this wide range, one should observe the Cu(II)|Cu(I) as well as Cu(I)|Cu(0). The authors have not shown the derivative plot. From the Fig 3a it seems there is a shallow peak between 3.25 to 3.5 V. This may be indeed the Cu(II)|Cu(I) redox couple and not the one (at 2.85 V) referred in the manuscript. It is strongly felt that both redox couples are operative in the RM(II)-BTC and the electrocatalysis involves both Cu(II)|Cu(I) as well as Cu(I)|Cu(0).

Response: We thank the reviewer for this valuable suggestion very much. According to previous literatures²¹⁻²³, the Cu(I)|Cu(0) redox potentials is in the region below 2.0 V vs. Li/Li⁺ (*Journal of Electroanalytical Chemistry*, 2019, 837, 76; *Electrochimica Acta*, 2022, 424, 140629; *ACS Catalysis*, 2023, 13, 12673). Obviously, the individual redox potentials of active metal centers in complex compounds are influenced by the properties and structure of ligands.

In order to identify the redox peaks of RM(II)-BTC more clearly, CV tests were presented. In Fig. R11 (Fig. 3a), the cyclic voltammetry curve of the solid RM(II)-BTC cathode in a CO₂ atmosphere (red curve) exhibits a reduction peak E_{c2} (2.81 V), corresponding to the electrochemical reduction process of RM(II)-BTC to Li₂RM(I)-BTC, which is the reaction equation R1 (equation 3 in revised manuscript) described in the paper. It's worth noting that, at this point, the reduction product Li₂RM(I)-BTC will be oxidized by the abundant CO₂ gas at the electrode interface, reverting back to RM(II)-BTC (as described in steps 2 and 3 in Fig. 5e). However, this oxidation process is a chemical reaction and not an electrochemical process involving electron transfer on circuit, so it doesn't appear as a corresponding peak on the cyclic voltammetry curve.

At this point, along with the electrode components reverting to RM(II)-BTC, there may still be a small amount of unoxidized residual $\text{Li}_2\text{RM(I)-BTC}$ and the reduction product of CO_2 , lithium oxalate. Looking at the anodic part of this cyclic voltammetry curve, you will notice an oxidation peak with somewhat poor symmetry. This oxidation peak is somewhat broad, and its left shoulder is somewhat prominent. This is a typical double-peak overlap phenomenon (E_{a2} and E_{a2}'). We can infer that the oxidation peak E_{a2} (3.40 V) corresponds to the electrochemical oxidation and lithium removal reaction of $\text{Li}_2\text{RM(I)-BTC}$, which is the reverse reaction of equation R1 (equation 3 in revised manuscript). This is similar to the oxidation reaction of the electrode in an Ar gas environment (E_{a1} peak in the green curve, 3.34 V). However, the oxidation peaks in Ar and CO_2 gas are different and shows a big shift in CO_2 with respect to Ar. This is **owing to the presence of CO_2 discharge products on the electrode surface with low electrical conductivity ($\text{Li}_2\text{C}_2\text{O}_4$), resulting in enhanced electrode polarization.** The shift between E_{a1} and E_{a2} corresponding to the oxidation peak of Cu(I) to Cu(II) is approximately 0.06 V, which is not very significant.

So, we believe that for the solid complex compound we proposed, **the reduction potential for the conversion between Cu(I) to Cu(0) should be below 2 V. Therefore, under our electrochemical window, this transition process cannot be observed.** Instead, there is an electrochemical transformation of Cu(II)|Cu(I) in the process of CO_2 electrochemical reduction.

Fig. R11 The CV plots of CNTs and solid RM(II)-BTC cathode in Ar and CO_2 atmosphere at 0.1 mV/s.

4. In the manuscript the discharge peak at 3.63 V has been referred to as due to the discharge products. This is very vague and speculative. The authors should demonstrate evidence to back up this statement.

Response: We thank the reviewer for this valuable suggestion very much. According to the measured CV curve Fig. R11 (Fig. 3a), there is one pair of redox peak in the CV curve under Ar and CO₂ atmosphere, corresponding to Cu(II)|Cu(I) redox peak in solid RM(II)-BTC. Although, the oxidation peaks in Ar and CO₂ gas are different and shows a big shift in CO₂ with respect to Ar. This is owing to the presence of carbon dioxide discharge products on the electrode surface with low electrical conductivity (Li₂C₂O₄), resulting in enhanced electrode polarization. Furthermore, there is also a sole oxidation peak at 3.70 V in CO₂, which does not exist in Ar, indicating that the oxidation peak is the result of the joint action of solid RM(II)-BTC and CO₂, rather than the result of the separate action of solid RM(II)-BTC material.

Fig. R12. (a) Raman, (b) FT-IR and (c) XRD spectrums of solid RM(II)-BTC cathode at different stages (pristine, discharge, recharge). (d) DEMS test during charging of Li-CO₂ battery using solid RM(II)-BTC cathode.

Regarding the evidence of the generation and decomposition process of lithium oxalate during discharge and charge processes, we have **designed a series of ex-situ and in-situ experiments**. These include non-in situ Raman spectroscopy, infrared

spectroscopy, and XRD experiments after discharge and charge, as well as in-situ mass spectrometry experiments during charging. We hope that these experimental studies can clarify the reviewers' concerns. The specific experimental results and analysis are as follows:

As depicted in Fig. R12a (Fig. 4a), compared to *ex-situ* Raman spectroscopy of the pristine RM(II)-BTC cathode, a small sharp peak indicating $\text{Li}_2\text{C}_2\text{O}_4$ at 1487 cm^{-1} is observed during the discharge process²⁴. Additionally, noticeable peaks at 605, 1417, 1560 and 1607 cm^{-1} , corresponding to $\rho(\text{O}-\text{C}=\text{O})$, $\delta_s(\text{C}-\text{O}) + \rho_\omega(\text{C}-\text{O})$, and $\nu_a(\text{O}-\text{C}=\text{O})$ of $\text{Li}_2\text{C}_2\text{O}_4$ respectively in Fig. R12b (Fig. 4b), are detected in the discharged cathode via FT-IR spectroscopy^{25,26}. As shown in Fig. R12c (Fig. 4c), besides several characteristic peaks of solid RM(II)-BTC, three main peaks centering at 19.78° , 27.39° , and 34.76° can be observed in the *ex-situ* XRD test, assigned to the (011), (-101), and (111) planes of standard $\text{Li}_2\text{C}_2\text{O}_4$ (JCPDS card No. 24-0646)^{27,28}. Subsequently, the charging result of DEMS shows that the charge-to-mass ratio is $1.05e^-/\text{CO}_2$ in Fig. R12d (Fig. 4f), which is close to the theoretical value of $1e^-/\text{CO}_2$ based on the reversible CO_2 evolution reaction in equation R2 (equation 4 in revised manuscript). The **DEMS experiment with a charging voltage plateau of 3.7 V, which is close to the potential value of the oxidation peak E_{a2}' in the CV experiment.** This further verifies that the primary charging reaction is the decomposition of $\text{Li}_2\text{C}_2\text{O}_4$ product.

Through these characterization techniques provides irrefutable evidence for the reversible formation and decomposition of crystalline $\text{Li}_2\text{C}_2\text{O}_4$ on solid RM(II)-BTC cathode, and the charging potential is just around 3.70V, corresponding to the oxidation peak of the new CV plot.

5. The statement “An inconspicuous slope appears at the start of the charging curve in the battery containing solid RM(II)-BTC, possibly due to the oxidation of Cu(I) to Cu(II) after deep discharge” is very vague. Why should this be due to a redox couple and not due to existence of multiphases? What is the evidence?

Response: We thank the reviewer for this valuable suggestion very much. To verify our conclusion, we performed an XPS test on a fully discharged cathode.

As shown in Fig. R13 (Supplementary Fig. 3), the deconvoluted Cu 2p spectrum of cathode after deep discharge shows peaks assigned to Cu(I) and Cu(II). From the XPS evidence, it can be seen that after deep discharge, there is $\text{Li}_2\text{RM(I)-BTC}$ containing monovalent copper within the electrode. These are produced during the electrochemical reduction process after deep discharge. During the charging process after discharge, this reduced state substance will be electrochemically oxidized to RM(II)-BTC . The specific capacity provided by this Li ion extraction process is only 91 mAh g^{-1} , as shown in Fig. R14 (Supplementary Fig. 4). This small specific capacity, when plotted on the x-axis of a $10,000 \text{ mAh g}^{-1}$ range, shows only a very slight slope. The charging curve after this slope corresponds to the electrochemical decomposition process of a large amount of discharge product, lithium oxalate.

As a consequence of this additional evidence, we get the following conclusion and corrected the related description as follows: “*the inconspicuous slope appears at the start of the charging curve in the battery containing solid RM(II)-BTC , possibly due to the oxidation of Cu(I) to Cu(II) after deep discharge. As shown in Supplementary Fig. 3, the deconvoluted Cu 2p spectrum of cathode after deep discharge shows peaks assigned to Cu(I) and Cu(II) ”.*

Fig. R13 XPS characterization of $\text{Cu 2p}_{3/2}$ for deep discharged Li-CO_2 battery based on RM(II)-BTC cathode.

Fig. R14. The first Charging/discharging curves of solid RM(II)-BTC cathode in Ar atmosphere at 100 mA g⁻¹.

6. Fig 3b: The CNT discharge potential is too low and is usually expected at 2.3 to 2.5 V. This is a very surprising observation.

Response: We thank the reviewer for this valuable comment. If the CNT electrode surface does not carry any catalyst, its catalytic activity for the electrochemical reduction of CO₂ to lithium carbonate and carbon is very low. This low catalytic activity becomes even more apparent in a high-purity carbon dioxide atmosphere. This experiment was conducted inside a CO₂ glovebox, where the oxygen and water vapor content were both below 0.01 ppm. After multiple repeated experiments, we indeed found that the catalytic activity of CNTs is very poor. A similar phenomenon has also been discovered and reported in a recent PNAS paper²⁹ (*PNAS*, 2023, 120, e2217454120). In this literature, the discharge curve of carbon materials in a high-purity carbon dioxide atmosphere even goes as low as 1.1 V.

7. The authors have demonstrated the Li₂RM(I)-BTC reaction with CO₂ and it's oxidization to Li₂RM(II)-BTC-2CO₂⁻ using a chemical experiment using CoCp. However, this may not exactly replicate the electrochemical reaction. The authors should demonstrate an in situ UV-VIS spectroscopy data to substantiate their claims.

Response: We thank the reviewer for this kind suggestion. This UV-visible

spectroscopy is based on the color change of CoCp in the reaction solution to determine the oxidation state changes of CoCp, thereby reducing the oxidation state of the added ions in the solution. The whole testing process is quite complicated and does not seem to be easily characterized by *in situ* UV-VIS spectroscopy.

At the request of the reviewer, we attempted to reconsider the in-situ experimental procedures. The specific experimental method may be as follows: First, the battery based on RM(II)-BTC cathode are discharged in Ar or CO₂ atmosphere, respectively. Then, the batteries are disassembled, the discharged cathodes are removed and cleaned with dry TEGDME solvent. Finally, the discharged cathode powder is added to TEGDME solution containing CoCp and tested by UV-VIS spectroscopy. If this in-situ UV-VIS spectroscopy test is to be carried out, CoCp needs to be added to the battery electrolyte, which is bound to affect the battery reaction. So, we sincerely apologize that we were unable to design an experiment for in-situ UV- VIS spectroscopy of battery reactions relying on color indicators.

Fortunately, these operations are carried out in the Ar glove box to guarantee that the valence state of copper ion is not changed by air. Even then, when UV-VIS tests are carried out in the environment, the reactions between CoCp and copper ion are completed. Therefore, the ex-situ UV-VIS spectroscopy can **also reveal the exact valence states of copper ion of RM(II)-BTC cathode** under different reaction states.

8. In Fig 5a, the authors show only the Cu XPS data. What are the changes in the Li1s and C1s data. These changes should be shown in the manuscript for a precise understanding of the reaction pathways and energy storage characteristics.

Response: We thank the reviewer for this valuable suggestion. To understand the reaction pathways exactly according to your suggestion, we have **supplemented the Li 1s and C 1s data during discharge in Fig. R15a-d (Fig.5a-d)**. Next, the formation mechanism of Li₂C₂O₄ during discharge will be discussed in detail.

Fig. R15 XPS Cu 2p_{3/2} spectra (a), Li 1s spectra (b) and C 1s spectra (c) of solid RM(II)-BTC cathode at different discharge stages.

First, through the XPS test, we find that the Cu 2p spectra of the pristine RM(II)-BTC cathode in Ar (Fig. R15a, top) is fitted into 2p_{3/2} peak at around 934.5 eV, indicating that only one valence state (Cu(II)) exists in pristine RM(II)-BTC cathode. As the discharge deepens (Fig. R15a, middle), a new peak at ~932.9 eV in Ar ascribed to the low valence state of Cu(I) rapidly increases, and the portion of Cu(II) is almost completely replaced by Cu(I). Meanwhile, a new peak at 55.89 eV is identified in the Li 1s spectra when compared to the pristine RM(II)-BTC cathode (Fig. R15b, top and middle). It may be related to the fact that the high valence state (RM(II)-BTC) has been partially reduced to the low valence state (Li₂RM(I)-BTC)¹⁰. By comparison, when CO₂ is introduced, the proportion of Cu(II) stays the same as the pristine stage (Fig. R15a, bottom), indicating that the reduced Li₂RM(I)-BTC may react with CO₂ and then be oxidized to Li₂RM(II)-BTC-2CO₂⁻ immediately. Subsequently, the characteristic peaks of O-C=O at 288.90 eV and Li at 55.34 eV for Li₂C₂O₄ are observed in the C 1s and Li 1s spectra, respectively¹¹ (Fig. R15c).

Fig. R16. (a) Raman, (b) FT-IR and (c) XRD spectrums of solid RM(II)-BTC cathode at different stages (pristine, discharge, recharge).

To provide further clarification that $\text{Li}_2\text{C}_2\text{O}_4$ is indeed produced by the interaction of reduced $\text{Li}_2\text{RM(I)-BTC}$ species and CO_2 via low valence state Cu(I) , we explored the Raman spectroscopy of solid RM(II)-BTC under different reaction conditions. For the pristine RM(II)-BTC, one pair of signal peak at $462/505\text{ cm}^{-1}$ referring to the vibration modes of Cu-O can be observed, respectively (Fig. R15d). The position of the Cu-O peak does not change when discharging in Ar, even though, at this point the high valence state Cu(II) has been reduced to the low valence state Cu(I) . The discharge process is then continued and CO_2 gas is injected. We observe that a couple of new peaks appearing at 482 and 520 cm^{-1} arise in the Raman spectroscopy, which is consistent with that of the bridged Cu-CO_2^- adduct¹². These results indicate that reduced species $\text{Li}_2\text{RM(I)-BTC}$ capture CO_2 to form the $[\text{Li}_2\text{RM(II)-BTC}]\text{-}2\text{CO}_2^-$ adduct primarily. Based on Raman spectroscopy, XRD spectroscopy and FT-IR techniques (Fig. R16), the discharge product on the surface of RM(II)-BTC cathode is $\text{Li}_2\text{C}_2\text{O}_4$.

Fig. R17 The local three-dimensional space-filling diagram of solid RM(II)-BTC cathode and

schematic illustration of reaction mechanism to form $\text{Li}_2\text{C}_2\text{O}_4$ during discharge of solid RM(II)-BTC cathode in the Li- CO_2 battery. The gray, red, cyan, green, and blue balls represent C, O, Cu(II), Cu(I) and Li^+ , respectively.

Hence, combined with the observed $\text{Li}_2\text{C}_2\text{O}_4$ product, we infer that **solid RM(II)-BTC catalyzes the discharge process** of Li- CO_2 battery, leading to the **generation of $\text{Li}_2\text{C}_2\text{O}_4$ discharge product** via a dimerization catalytic reaction, which is shown in Fig. R17. The whole discharge reaction mechanism is divided into four steps. Among these steps, step 1 involves an electrochemical reaction, while steps 2, 3, and 4 involve chemical reactions that have negligible reaction time. During the discharge process, the high valence RM(II)-BTC is electrochemically reduced, resulting in the production of a low valence state $\text{Li}_2\text{RM(I)-BTC}$ in step 1. This reduction is achieved through the insertion of lithium ions into the RM(II)-BTC lattice. After step 2, the absorbed CO_2 is captured by the reduced $\text{Li}_2\text{RM(I)-BTC}$ to form the bridging unit (Cu(II)-CO_2^-) through a chemical reaction. Then, in step 3, two bridging units (Cu(II)-CO_2^-) polymerize into a dimeric oxalate intermediate ($\text{Cu(II)-C}_2\text{O}_4^{2-}\text{-Cu(II)}$). In step 4, Li ions combine with the dimeric oxalate intermediate, evolving $\text{Li}_2\text{C}_2\text{O}_4$ product and dissociating from the solid RM(II)-BTC. The discharge process continues as these cycles (four steps) repeat again, with an increasing discharge capacity.

9. The dimer model is primarily shown using DFT studies. There are practically no experimental support for this proposition remains inconclusive. There are several questions here. What is the evidence of LiCO_2^- being anchored to the RM(II)-BTC and that the eventual formation of $\text{Li}_2\text{C}_2\text{O}_4$ being anchored to it. This can also be in the solution very close to the porous electrode.

Response: We thank the reviewer for this valuable comment. We can provide some evidence of the existence of LiCO_2^- and offer a detailed explanation. During the discharge process, we observe that a couple of new peaks appearing at 482 and 520 cm^{-1} arise in the Raman spectroscopy, which is consistent with that of the bridged Cu-CO_2^- adduct¹². Furthermore, noticeable peaks at 605, 1417, 1560 and 1607 cm^{-1} ,

corresponding to $\rho(\text{O}-\text{C}=\text{O})$, $\delta_s(\text{C}-\text{O}) + \rho_w(\text{C}-\text{O})$, and $\nu_a(\text{O}-\text{C}=\text{O})$ in $\text{C}_2\text{O}_4^{2-}$ group are detected in the discharged cathode. The characteristic peak of $\text{O}-\text{C}=\text{O}$ for $\text{C}_2\text{O}_4^{2-}$ group is detected at 288.95 eV in the C 1 s spectrum through the newly tested XPS spectra (**Fig. R15c**). In fact, these data are sufficient to demonstrate the existence of dimerization reactions.

In this manuscript, we did not emphasize that LiCO_2^- is anchored to RM(II)-BTC and that the eventual formation of $\text{Li}_2\text{C}_2\text{O}_4$ being anchored to it. We emphasize the role of RM(II)-BTC in the electrochemical reduction of CO_2 . During the discharge process, the high valence RM(II)-BTC is electrochemically reduced, resulting in the production of a low valence state $\text{Li}_2\text{RM(I)-BTC}$ in step 1. This reduction is achieved through the insertion of lithium ions into the RM(II)-BTC lattice. After step 2, the absorbed CO_2 is captured by the reduced $\text{Li}_2\text{RM(I)-BTC}$ to form the bridging unit (Cu(II)-CO_2^-) through a chemical reaction. Then, in step 3, two bridging units (Cu(II)-CO_2^-) polymerize into a dimeric oxalate intermediate ($\text{Cu(II)-C}_2\text{O}_4^{2-}\text{-Cu(II)}$). In step 4, Li ions combine with the dimeric oxalate intermediate, evolving $\text{Li}_2\text{C}_2\text{O}_4$ product and dissociating from the solid RM(II)-BTC .

Finally, we know that the reaction gas (CO_2) diffuses to the electrolyte and the surface of the porous electrode during the discharge process, and the reduction reaction occurs at the gas-liquid-solid three-phase interface. It seems difficult to determine whether the reaction occurs on the RM(II)-BTC surface or in a solution close to the porous electrode. We think that these two conditions should coexist. However, the highly porous complex compound material with a high surface area that we have provided as a solid-state redox mediator for lithium- CO_2 batteries can adsorb a greater number of CO_2 molecules^{30,31} (*Advanced Functional Materials* 2014, 24, 3855-3865; *Chemical Reviews* 2017, 117, 14, 9674-9754). Therefore, we believe that carbon dioxide molecules reacting on the surface of RM should dominate.

10. Supplementary Information Figure 4 mentions in situ Raman however the data shows XRD. The error needs to be rectified.

Response: We were really sorry for our careless mistake. Thank you for your reminder. We have corrected the “In-situ Raman spectra” into “In-situ XRD spectra”.

11. What is the basis for the selection of the EIS circuit shown in Supplementary Information Figure 6? It does not seem appropriate. What are the error bars and what are the capacitance values?

Response: We thank the reviewer for this valuable comment. Through literature research^{32,33} (*Electrochemistry Communications*, 2013, 34, 77-80; *Electrochemical and Solid-State Letters*, 2010, 13, 121-124), we found that the original EIS circuit is not appropriate, and then we **put forward a new equivalent EIS circuit (Fig. R18)**.

The circuit elements are defined as follows, R_s reflects the ohmic resistance that comprises contributions from the electrolyte, electrodes, current leads, and so on. R_{sei} denotes the resistance of solid electrolyte interfacial layers on the air electrode, whereas C_{sei} represents its capacitance. Similarly, R_{ct} and C_{dl} are attributable to charge-transfer resistance and double-layer capacitance, respectively. W represents the Warburg impedance, that arises from a diffusion process. At the same time, we have **added the error bars and the capacitance values to Table R1 below** (Supplementary Table 1 in revised manuscript).

Fig. R18 Equivalent circuit model of Li-O₂ battery.

Table R1 Circuit values with different catalysts at a fixed capacity (1000 mAh g⁻¹).

Catalysts	Current	R_t (Ω)	R_s (Ω)	R_{int} (Ω)	C_{int} (μ F)	C_{dl} (μ F)	Error (%)
CNTs	1000 mA g ⁻¹	670.5	15.6	22.6	0.34	18.2	4.76
	1500 mA g ⁻¹	965.2	17.2	23.7	0.36	19.3	4.68
Soluble RM	1000mA g ⁻¹	260.6	48.5	21.3	0.29	14.4	4.53

	1500mA g ⁻¹	420.7	50.2	22.4	0.31	15.1	4.69
RM(II)-BTC	1000mA g ⁻¹	110.4	14.8	21.1	0.27	7.3	4.43
	1500mA g ⁻¹	140.8	15.6	22.5	0.29	7.9	4.64

12. Generally the manuscript figures are poor quality. Figure 1a is more a figure suited for a review paper and is not appropriate here. This should be removed from Figure 1. Many of the other panels should be reduced. Additionally, Figure 1 should be reorganized for better clarity. Similarly, Figure 3g is redundant as the authors put up a comparison Table in the ESI. This should also be removed from Fig 3.

Response: We thank the reviewer for this valuable suggestion. As suggested by the reviewer, we have removed the Figure 1a from the Figure 1 and reorganized the Figure 1 (Fig. R19) for better clarity. Meanwhile, we also have removed the Figure 3g from the Figure 3 and added it in Supplementary Fig. 10.

Fig. R19. (a) Schematic of Li-CO₂ battery with solid RM(II)-BTC cathode. RM(II)-BTC can inhibit the shuttle effect since it is fixed at the cathode as a solid state and exhibit high CO₂ adsorption, fast kinetics and reversible redox couples. (b) Solid RM(II)-BTC-mediated reaction mechanism towards Li₂C₂O₄ route. RM(II)-BTC is electrochemically reduced to RM(I)-BTC, which immediately reacts with CO₂ to form RM(II)-BTC-C₂O₄²⁻. Finally, discharge product Li₂C₂O₄ and origin RM(II)-BTC are obtained. The frame with dash-dotted line reflect that after receiving electrons and incorporating lithium ions, RM(II)-BTC can transfer to a compound of lithium and RM(I)-BTC. The gray, red, cyan, green, and blue balls represent C, O, Cu(II), Cu(I) and Li⁺, respectively.

Reviewer #3 (Remarks to the Author):

The study of the electrochemical reduction of carbon dioxide in organic electrolytes has not only significant scientific importance but also practical significance for achieving carbon peak and carbon neutrality goals. I believe that this article makes an important contribution to the field of lithium-carbon dioxide batteries. In the past, the redox mediators used were soluble materials, and these soluble redox mediators improved the energy storage efficiency of the batteries by altering the reduction and deposition pathways of carbon dioxide, reducing the electrochemical overpotential. The proposed solid-state redox mediator in this paper presents a promising approach. It not only avoids the shuttle effect associated with liquid redox mediators but also enhances the reaction rate. The evidence presented in the paper supports the proposed reaction pathway where carbon dioxide molecules, assisted by copper complexes, form oxalate ions. I believe that this innovative approach will attract widespread attention from researchers in the field of metal-air batteries and open a new realm of solid-state redox mediator design research for lithium-carbon dioxide batteries and even metal-air batteries. Therefore, I recommend it for publication in Nature Communications after the following revisions.

Response: We thank the reviewer for this positive comment on the importance of our work. Please see the point-to-point responses below.

1. The sentence “the charging result of DEMS shows that the charge-to-mass ratio is $0.95e^-/CO_2$, which is close to the theoretical value of $1e^-/CO_2$ based on the reversible CO_2 evolution reaction (Equation 4)” The authors should explain this point and provide the calculation process of the charge-to-mass ratio ($0.95e^-/O_2$) in greater detail.

Response: We thank the reviewer for this suggestion very much. Since $Li_2C_2O_4$ is identified as the discharge product through a series of spectroscopic characterizations, it is reasonable to speculate that the charge electrochemical process is as follows:

And the theoretical charge-to-mass ratio should be $1e^-/\text{CO}_2$. In-situ DEMS was conducted to verify the speculation by measuring transferred electrons and generated CO_2 during the charge process. The practical value of electrons (n_e) based on the charge capacity (Q_{the}) can be calculated as follows:

$$Q_{the} = It = 0.15 \text{ mA} \times 10 \text{ h} = 1.5 \text{ mAh} = 1.5 \times 10^{-3} \text{ A} \times 3600 \text{ s} = 5.4 \text{ C}$$

$$n_e = \frac{Q_{the}}{Q_e \times N_A} = \frac{5.4 \text{ C}}{1.602 \times 10^{-19} \text{ C} \times 6.02 \times 10^{23} \text{ mol}^{-1}} = 55.99 \text{ } \mu\text{mol}$$

$$\frac{Z}{m} = \frac{n_e}{n_{\text{CO}_2}} = \frac{55.99 \text{ } \mu\text{mol}}{53.32 \text{ } \mu\text{mol}} = 1.05$$

Based on the CO_2 evolution quantity (n_{CO_2} , 53.32 μmol), the practical charge-to-mass ratio is 1.05 close to $1e^-/\text{CO}_2$, confirming the CO_2 evolution based on equation $\text{Li}_2\text{C}_2\text{O}_4 \rightarrow 2\text{Li}^+ + 2e^- + 2\text{CO}_2$.

However, we obtained 0.95 e^-/CO_2 in the manuscript because I used the wrong formula as follow:

$$\frac{Z}{m} = \frac{n_{\text{CO}_2}}{n_e} = \frac{53.32 \text{ } \mu\text{mol}}{55.99 \text{ } \mu\text{mol}} = 0.95$$

We apologized for making this mistake and revised the results (1.05 e^-/CO_2) in revised manuscript. Meanwhile, we have added the calculation process of the practical charge-to-mass ratio in detail in the supplementary information of revised manuscript.

2. To prove that the CO_2 is indeed involved in the reaction, please supply the DEMS data of discharging process.

Response: Obviously the reviewer has made a very meaningful suggestion. To prove that the CO_2 is indeed involved in the discharge reaction, we have carried out in situ DEMS measurement to monitor CO_2 consumption during the discharge process. As the discharge progresses, CO_2 is continuously consumed in Fig. R20. At the same time, we added relevant descriptions in revised manuscript (Supplementary Fig. 15).

Fig. R20 DEMS test during discharging of Li-CO₂ battery using solid RM(II)-BTC cathode.

3. What is the mass loading of the active material of the flexible pouch-type Li-CO₂ battery?

Response: We thank the reviewer for this suggestion very much. The mass loading of the active material is 10 mg in the flexible pouch-type Li-CO₂ battery (5 cm × 8 cm). And we have added it in the supplementary information of revised manuscript.

4. Aside from the electrochemical stability, the stability of solid RM(II)-BTC after cycles are also essential to Li-CO₂ batteries, therefore, the related stability is required to be supplemented.

Response: According to the reviewer's suggestion, we have performed XRD test of RM(II)-BTC cathode to manifest the stability after cycles. The following XRD measurements illustrate that all peaks assigned to solid RM(II)-BTC and is consist with the initial structure after 400 cycles (Fig. R21). We have corrected the related description in revised manuscript and added the XRD measurement of RM(II)-BTC cathode after 400 cycles in the supplementary information (Supplementary Fig. 9 in revised manuscript).

Fig. R21 XRD results of RM(II)-BTC cathode at pristine and 400 cycles. (**Supplementary Fig. 9**)

Some minor mistakes and typos:

1) The sentence “These results indicate that reduced species $\text{Li}_2\text{RM(I)-BTC}$ capture CO_2 to form the $[\text{Li}_2\text{RM(II)-BTC}]\text{-}2\text{CO}_2^{2-}$ adduct primarily”. The $[\text{Li}_2\text{RM(II)-BTC}]\text{-}2\text{CO}_2^{2-}$ adduct should be written to $[\text{Li}_2\text{RM(II)-BTC}]\text{-}2\text{CO}_2^-$.

Response: We thank the reviewer for this suggestion very much. We have revised the molecular formula in revised manuscript.

2) The format of physical units, such as S cm^{-1} , S/cm and mAh g^{-1} in this manuscript, needs to be unified.

Response: We thank the reviewer for this suggestion very much. We have unified the format of physical units in revised manuscript.

References

1. P. G. Bruce, et al. A stable cathode for the aprotic Li-O₂ battery. *Nature Materials* **12**, 1050-1056 (2013).
2. P. G. Bruce, et al. A rechargeable lithium-oxygen battery with dual mediators stabilizing the carbon cathode. *Nature Energy* **2**, 17118 (2017).
3. L. B. Hu, et al. Flexible lithium-CO₂ battery with ultrahigh capacity and stable cycling. *Energy & Environment Science* **11**, 3231-3237 (2018).
4. B. Wang, et al. Monodispersed MnO nanoparticles in graphene-an interconnected N-doped 3D carbon framework as a highly efficient gas cathode in Li-CO₂ batteries. *Energy & Environment Science* **12**, 1046-1054 (2019).
5. L. Song, C. G. Hu, Y. Xiao, J. P. He, Y. Lin, J. W. Connell, L. M. Dai, An Ultra-Long Life, High-Performance, Flexible Li-CO₂ Battery Based on Multifunctional Carbon Electrocatalysts. *Nano Energy* **71**, 104595 (2020).
6. Y. C. Jin, C. G. Hu, Q. B. Dai, Y. Xiao, Y. Lin, J. W. Connell, F. Y. Chen, L. M. Dai, High-Performance Li-CO₂ Batteries Based on Metal-Free Carbon Quantum Dot/Holey Graphene Composite Catalysts. *Advanced Functional Materials* **28**, 1804630 (2018).
7. S. J. Guo, et al. Biaxially Compressive Strain in Ni/Ru Core/Shell Nanoplates Boosts Li-CO₂ Batteries. *Advanced Materials* **34**, 2204134 (2022).
8. Y. G. Li, et al. Conjugated Cobalt Polyphthalocyanine as the Elastic and Reprocessable Catalyst for Flexible Li-CO₂ Batteries. *Advanced Materials* **31**, 1805484 (2019).
9. J. Chen, et al. Mo₂C/CNT: An Efficient Catalyst for Rechargeable Li-CO₂ Batteries. *Advanced Functional Materials* **27**, 1700564 (2017).
10. T. M. Ivanova, K. I. Maslakov, A. A. Sidorov, M. A. Kiskin, R. V. Linko, S. V. Savilov, V. V. Lunin, I. L. Eremenko, XPS detection of unusual Cu(II) to Cu(I) transition on the surface of complexes with redox-active ligands. *Journal of Electron Spectroscopy and Related Phenomena* **238**, 146878 (2020).
11. N. Feng, et al. Mechanism-of-action elucidation of reversible Li-CO₂ batteries using the water-in-salt electrolyte. *ACS Applied Materials & Interfaces* **13**, 7396-7404 (2021).
12. W. Zhang, et al. Dynamic Restructuring of Coordinatively Unsaturated Copper Paddle Wheel Clusters to Boost Electrochemical CO₂ Reduction to Hydrocarbons. *Angewandte Chemie International Edition* **61**, e202112116 (2022).
13. H. S. Zhou, et al. Towards a stable Li-CO₂ battery: The effects of CO₂ to the Li metal anode. *Energy Storage Materials* **26**, 443-447 (2020).
14. H. S. Zhou, et al. Binuclear Cu complex catalysis enabling Li-CO₂ battery with a high discharge voltage above 3.0 V. *Nature Communications* **14**, 536 (2023).
15. P. G. Bruce, et al. Promoting solution phase discharge in Li-O₂ batteries containing weakly solvating electrolyte solutions. *Nature Materials* **15**, 882-888 (2016).
16. Z. Zhang, et al. Enhanced electrochemical performance of aprotic Li-CO₂ batteries with a ruthenium-complex-based mobile catalyst. *Angewandte Chemie International Edition* **60**, 16404-16408 (2021).

17. D. Cao, X. Liu, X. Yuan, F. Yu, Y. Chen. Redox mediator-enhanced performance and generation of singlet oxygen in Li-CO₂ batteries. *ACS Applied Materials & Interfaces* **13**, 39341-39346 (2021).
18. X. Mu, H. Pan, P. He, H. Zhou. Li-CO₂ and Na-CO₂ batteries: toward greener and sustainable electrical energy storage. *Advanced Materials* **32**, 1903790 (2019).
19. D.Q. Cao, et al. Threshold potentials for fast kinetics during mediated redox catalysis of insulators in Li-O₂ and Li-S batteries. *Nature Catalysis* **5**, 193-201 (2022).
20. H.S. Zhou. Kinetics of redox-mediated catalysis in batteries. *Nature Catalysis* **5**, 173-174 (2022).
21. E. Chiyindiko, et al. Redox behaviour of bis(β-diketonato)copper(II) complexes. *Journal of Electroanalytical Chemistry* **837**, 76 (2019).
22. E. Chiyindiko, et al. Electrochemical behaviour of copper(II) complexes containing 2-hydroxyphenones. *Electrochimica Acta* **424**, 140629 (2022).
23. J. L. Martinez, et al. Electrochemical Reduction of N₂O with a Molecular Copper Catalyst. *ACS Catalysis* **13**, 12673 (2023).
24. H. G. M. Edwards, I. R. Lewis, FT-Raman spectroscopic studies of metal oxalates and their mixtures. *Spectrochimica Acta Part A: Molecular Spectroscopy* **50**, 1891-1898 (1994).
25. R. J. H. Clark, S. Firth, Raman, infrared and force field studies of K₂C₂O₄ · H₂O and K₂C₂O₄ · H₂O in the solid state and in aqueous solution, and of (NH₄)₂C₂O₄ · H₂O and (NH₄)₂C₂O₄ · H₂O in the solid state. *Spectrochimica Acta Part A: Molecular and Biomolecular Spectroscopy* **58**, 1731-1746 (2002).
26. A. R. Hind, et al. On the Aqueous Vibrational Spectra of Alkali Metal Oxalates. *Applied Spectroscopy* **52**, 683-691 (1998).
27. B. L. Shen, H. J. Zhang, Y. J. Wu, H. Jiang, Y. J. Hu, C. Z. Li, Co₃O₄ Quantum Dot-Catalyzed Lithium Oxalate as a Capacity and Cycle-Life Enhancer in Lithium-Ion Full Cells. *ACS Applied Energy Materials* **5**, 2112-2120 (2022).
28. D. Li, F. Lian, K. C. Chou, Decomposition mechanisms and non-isothermal kinetics of LiHC₂O₄ · H₂O. *Rare Metals* **31**, 615-620 (2012).
29. P. Tan, et al. Unveiling the mysteries of operating voltages of lithium-carbon dioxide batteries. *Proceedings of the National Academy of Sciences* **120**, e2217454120 (2023).
30. P. R. Javier, et al. Solvent-Mediated Reconstruction of the Metal-Organic Framework HKUST-1 (Cu₃(BTC)₂). *Advanced Functional Materials* **24**, 3855-3865 (2014).
31. P. B. Balbuena, et al. CO₂ Capture and Separations Using MOFs: Computational and Experimental Studies. *Chemical Reviews* **117**, 14, 9674-9754 (2017).
32. J. P. Zheng, et al. Investigation of a Li-O₂ cell featuring a binder-free cathode via impedance spectroscopy and equivalent circuit model analysis. *Electrochemistry Communications* **34**, 77-80 (2013).
33. L. G. Scanlon, et al. High Capacity Li-O₂ Cell and Electrochemical Impedance. *Spectroscopy Study Electrochemical and Solid-State Letters* **13**, 121-124 (2010).

REVIEWERS' COMMENTS

Reviewer #2 (Remarks to the Author):

The authors have revised the manuscript along the lines of the reviewers' comments. The manuscript can be accepted for publication in Nature Communications

Reviewer #4 (Remarks to the Author):

After reviewing the point-by-point response, I believe the concerns have been adequately addressed. Therefore, I support its publication at present form.

Response to Reviewers' Comments

We greatly appreciate all reviewers for their constructive comments and suggestions, which has in our view significantly raised the quality of the manuscript (NCOMMS-23-24308A). The responses to the reviewers' comments are as follows.

Point-to-Point Responses (see next page)

Reviewer #2 (Remarks to the Author):

The authors have revised the manuscript along the lines of the reviewers' comments. The manuscript can be accepted for publication in Nature Communications.

Response: We thank the reviewer for the positive comment to our work.

Reviewer #4 (Remarks to the Author):

After reviewing the point-by-point response, I believe the concerns have been adequately addressed. Therefore, I support its publication at present form.

Response: We thank the reviewer for the positive comment to our work.